
# Impacts of spatio-temporal resolutions of precipitation on flood events simulation based on multi-model structures — A case study over Xiang River Basin in China

Qian Zhu[1, *], Xiaodong Qin[1], Dongyang Zhou[1], Tiantian Yang[2], Xinyi Song[3]

[1]School of Civil Engineering, Southeast University, Nanjing 211189, China; zhuqian@seu.edu.cn

[2]School of Civil Engineering and Environmental Science, University of Oklahoma, Norman, OK, 73019 , USA; Tiantian.Yang@ou.edu

[3]School of Hydraulic and Environmental Engineering, Changsha University of Science & Technology, Changsha 410114, China; songxy@csust.edu.cn

*Correspondence to*: Qian Zhu(zhuqian@seu.edu.cn)

**Key words:** satellite-based precipitation; spatio-temporal resolutions; hydrological modeling; Long Short-Term Memory (LSTM); calibration strategies; flood events

**Abstract.** Accurate flood events simulation and prediction, enabled by effective models and reliable data, are critical for mitigating the potential risk of flood disaster. This study aims to investigate the impacts

of spatio-temporal resolutions of precipitation on flood events simulation in a large-scale catchment of China. We use the high spatio-temporal resolutions Integrated Multi-satellite Retrievals for Global Precipitation Measurement (IMERG) products and a gauge-based product as precipitation forcings for hydrologic simulation. Three hydrological models (HBV, SWAT, and DHSVM) and a data-driven model (Long Short-Term Memory (LSTM) network) are utilized for flood events simulation. Two calibration

strategies are carried out, one of which targets at matching the flood events and the other one is the conventional strategy to match continuous streamflow. The results indicate that the event-based calibration strategy improves the performance of flood events simulation, compared with conventional calibration strategy, except for DHSVM. Both hydrological models and LSTM yield better flood events simulation at finer temporal resolution, especially in flood peaks simulation. Furthermore, SWAT and

DHSVM are less sensitive to the spatial resolutions of IMERG, while the performance of LSTM obtains improvement when degrading the spatial resolution of IMERG-L. Generally, the LSTM outperforms the hydrological models in most flood events, which implies the usefulness of the deep learning algorithms for flood events simulation.

## 1 Introduction

The global climate change increases the risk of floods, which brings heavy casualties and losses of

property (Hirabayashi et al. 2013). In China, the flood events seem to become more frequent over the

mid to lower reaches of the Yangtze River due to the increasing intensity and frequency of rainfall

extremes (Piao et al. 2010). In June 2017, large-scale flood events induced by heavy rainfall in Hunan



province, located in the southern China, affected more than ten million people and caused economic

losses of more than 40 billion Chinese Yuan. Reliable flood events simulation and prediction are the key

to minimize the losses and impacts caused by flood events.

Numerous models are applied to simulate the flood events, most of which are physically based hydrologic

models (Dutta et al. 2000, Koutroulis and Tsanis 2010, Nikolopoulos et al. 2013, Wu et al. 2014, Mei et

al. 2016, Yang et al. 2017, Yu et al. 2018, Grimaldi et al. 2019), and others are based on artificial neural

networks (Shrestha et al. 2005, Badrzadeh et al. 2015). Owing to the continuous development of artificial

neural networks, deep learning (DL) is emerged as a dominant tool, which impacts various scientific

disciplines in recent years (Akbari Asanjan et al. 2018, Shen 2018, Shen et al. 2018, Zhang et al. 2018).

Among various DL methods, the Long Short-Term Memory (LSTM) network is appropriate for

capturing the relationship between rainfall and runoff, because of its ability to learn long-term

dependencies and delays between the input and output, and shows extraordinary potential in hydrological

simulation (Hu et al. 2018, Liao et al. 2019, Fan et al. 2020, Kao et al. 2020, Ni et al. 2020, Zhu et al.

2020b). Both hydrological models and deep-learning based models require multi-sources inputs,

particularly precipitation, which is the key forcing variable in hyrological process to simulate/predict

flood events.

Traditionally, in-situ precipitation is utilized for hydrological simulation. However, because of the

uneven distribution of in-situ observations and its unavailability in less developed regions, satellite-based

precipitation products have been widely used as an alternative precipitation source, and further applied

for flood events simulation (Maggioni and Massari 2018). Among them, the Integrated Multi-satellite

Retrievals for Global Precipitation Measurement (IMERG) (Huffman et al. 2015) is a high spatio-

temporal resolution satellite-based precipitation product released by National Aeronautics and Space

Administration (NASA), whose accuracy and hydrological utility have been evaluated from multiple

aspects, such as with different temporal scales (e.g., daily and sub-daily) (Tang et al. 2016, Yuan et al.

2018, Su et al. 2020), and on basins with different climate conditions (O et al. 2017, Wang et al. 2017,

Zubieta et al. 2017, Fang et al. 2019, Jiang and Bauer-Gottwein 2019). Many studies show that the

performance of IMERG varies across different climate regions and terrain. In addition, most of the

IMERG-related studies are conducted to assess its performance at a specific spatio-temporal resolution,

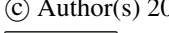



a few of which consider the impacts of different spatio-temporal resolutions on its accuracy. Among limited studies, Tang et al. (2016) evaluated the IMERG products at hourly, 3-hourly and daily scales, and they revealed that the statistical indices of IMERG increase with coarser temporal resolutions. Su et

al. (2020) assessed the IMERG products at multiple spatial and temporal resolutions by upscaling, and the study summarized that degrading the spatio-temporal resolution improves the accuracy of IMERG products. However, these two studies just evaluated the accuracy of IMERG products at multiple spatio-temporal scales, rather than the effects of spatio-temporal resolutions of IMERG products on their hydrological applications (e.g., flood simulation).

As proved by Huang et al. (2019), the spatio-temporal resolutions affect the accuracy of precipitation estimates, and the effects can be propagated to the flood events simulation through the hydrological processes. However, the impact of precipitation with different spatio-temporal resolutions on hydrological simulation has not yet been determined, which is related to many different factors, such as the structure of hydrological models (Arnaud et al. 2011, Yu et al. 2014), the scale of catchment and

event characteristics (Lobligeois et al. 2014, Ficchì et al. 2016). Most studies investigated the sensitivity of hydrological models to spatio-temporal resolution based on one model structure with in-situ precipitation, and they concluded that the accuracy of hydrological simulation is not always higher with shorter time steps or higher spatial resolutions (Liang et al. 2004, Arnaud et al. 2011, Lobligeois et al. 2014, Yu et al. 2014, Rafieeinasab et al. 2015, Ficchì et al. 2016, Melsen et al. 2016, Buitink et al. 2019,

Huang et al. 2019). For instance, some studies present better hydrological simulation forced by in-situ precipitation with lower spatio-temporal resolutions to some extent (Liu et al. 2012, Apip et al. 2012, Lobligeois et al. 2014, Ficchì et al. 2016). As we all know, high spatio-temporal resolution is one of the advantages of satellite-based precipitation products, however, there are also studies pointing out that degrading the spatio-temporal resolution can improve the accuracy of precipitation (Su et al. 2020). But

rare studies have been conducted to probe the effects of spatio-temporal of satellite-based precipitation on flood simulation, not to mention its impact on flood simulation with models based on DL methods (e.g., LSTM). What's more important, to our best knowledge, the sensitivity of models with different structures, such as lumped hydrological model, semi-distributed/distributed hydrological model, and data-driven model, to the spatio-temporal resolutions of precipitation has not been investigated.



Therefore, three widely used and typical physically-based models (lumped HBV model, semi-distributed SWAT model, and distributed DHSVM model), and one data-driven model (LSTM) which shows good performance in hydrological simulation, are employed to probe the impacts of spatio-temporal resolutions of precipitation on flood events simulation.

Apart from the factors mentioned above, the rationality of calibration is another important factor to affect
the accuracy of hydrological simulation. Many studies investigate the influences of the choice of objective function and calibration method on hydrological simulation, but most of them use the calibration strategy based on continuous streamflow instead of flood events (Moussa and Chahinian 2009, Noilhan et al. 2010, Nikolopoulos et al. 2013, Badrzadeh et al. 2015, Yoshimoto and Amarnath 2017, Spellman et al. 2018). However, some studies prove that the event-based calibration can improve the
performance of streamflow simulation. For instance, Yu et al. (2018) developed the sub-daily SWAT-EVENT model for event-based flood simulation, which particularly improved the performance of flood events simulation, especially the accuracies of the flood peaks. And Xie et al. (2019) compared the continuous modeling and event-based modeling based on the generalized likelihood uncertainty estimation (GLUE), and found the event-based simulation showed better overall performance. However,
studies about event-based calibration is still quite limited, particularly for LSTM. Therefore, in this study, we conduct different calibration strategies aimed at obtaining the best possible flood events simulation.

The main objectives of this study are: (1) to investigate the impact of spatio-temporal resolutions of satellite-based precipitation estimates derived from IMERG on streamflow simulation, particularly flood events simulation, over a watershed of 82,375 km$^2$; (2) to explore and compare the performance of
hydrological models with different structures and LSTM on flood events simulation based on gauge-based and satellite-based precipitation products; (3) to study the potential benefits of the calibration strategy based on flood events. The remaining sections of the paper are organized as follows: the descriptions of the study area and data are presented in Section 2; the methodology is introduced in Section 3; Section 4 provides the results; the discussion is stated in Section 5; and conclusions are
summarized in Section 6.


## 2 Study area and data

### 2.1 Study area

The Xiang River basin is a humid region, located in the middle reach of the Yangtze River, within

110.50°E-114.25°E, 24.50°N-28.25°N in the southern China, which covers an area of about 82,375 km²

above the Xiangtan hydrological station (Fig. 1). Together with the impact of diverse topographic types

and a dominant subtropical monsoon climate, the precipitation is characterized with strong temporal and

spatial variability (Zhu et al. 2017). The average annual temperature of the basin is around 17°C, and the

mean annual total precipitation is around 1,400-1,700 mm, most of which falls from April to September.

Concentrated storm events during the flood season cause frequent floods throughout the basin, while the

Xiang River basin is the most densely populated and economically developed area in Hunan Province

(Zhu et al. 2020a). So, it is meaningful to simulate and predict flood events accurately over the study

area for flood risk management.

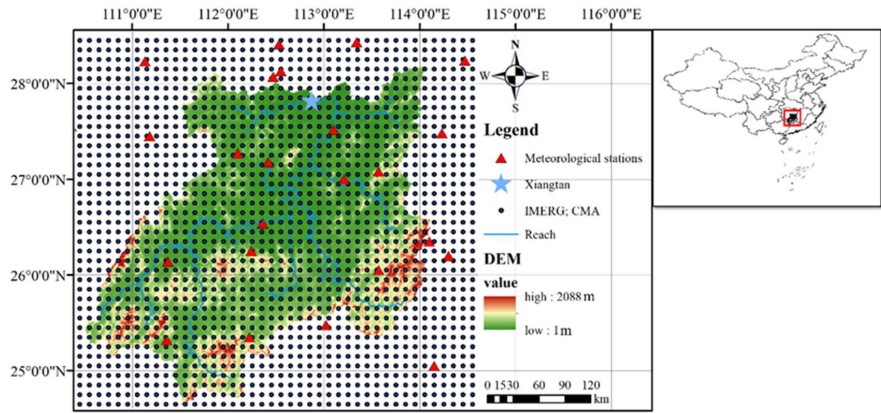

**Fig. 1. The spatial distribution of meteorological stations, the outlet of the study area, and precipitation from**
**IMERG and CMA.**

### 2.2 Data description

IMERG V05B is a widely used satellite-based precipitation product with a spatio-temporal resolution of

0.1° and 30-min released by NASA, which consists of multiple rainfall retrieval algorithms and combines

various precipitation-relevant remote sensing data sources obtained from the GPM sensors (Huffman et

al. 2015). The IMERG system is firstly run twice to produce IMERG Early Run and IMERG Late Run





(hereafter IMERG-E and IMERG-L) with a latency of 4 hours and 12 hours in near real-time (NRT). And then through the bias adjustment with monthly Global Precipitation Climatology Centre (GPCC) gauge observations, IMERG Final Run (hereafter IMERG-F) is generated with 2.5 months latency.

A precipitation product (hereafter CMA) released by China Meteorological Administration, which

merges rain gauge data from more than 30,000 automatic weather stations (AWSs) in China with the Climate Prediction Center morphing technique (CMORPH) precipitation product by an improved probability density function-optimal interpolation method (PDF-OI), is used as the reference precipitation dataset in this study (Shen et al. 2014). CMA provides precipitation estimates in a spatial resolution of 0.1° and a temporal resolution of 1-hour, which is proved to be a reliable precipitation

product as a result of high density of the AWSs and rigorous quality control of the source data. Therefore, CMA has already been applied as the benchmark in some studies (Wang et al. 2017, Tang et al. 2017, Su et al. 2020).

Daily gauge meteorological variables (maximum and minimum temperature, relative humidity, wind speed and solar radiation) at 27 meteorological stations over the Xiang River basin are obtained from

CMA. The available hourly streamflow observation at Xiangtan Station is provided by Hunan Hydrological Bureau of China. Fig. 2 shows the time series of the hourly streamflow and corresponding gauge-based precipitation between 2015 and 2017, where eleven historical flood events are selected with flood peak exceeding the threshold of 8,600 m3/s in this study. The period of the time series containing the selected flood events is from April 2014 to December 2017. The DEM (digital elevation model) with

90m resolution is derived from NASA Shuttle Radar Topographic Mission (SRTM) (Farr et al. 2007). Land cover and soil data with a resolution of 1km are obtained from Global Land Cover 2000 and Environmental and Ecological Science Data Center for West China, National Natural Science Foundation of China, respectively.





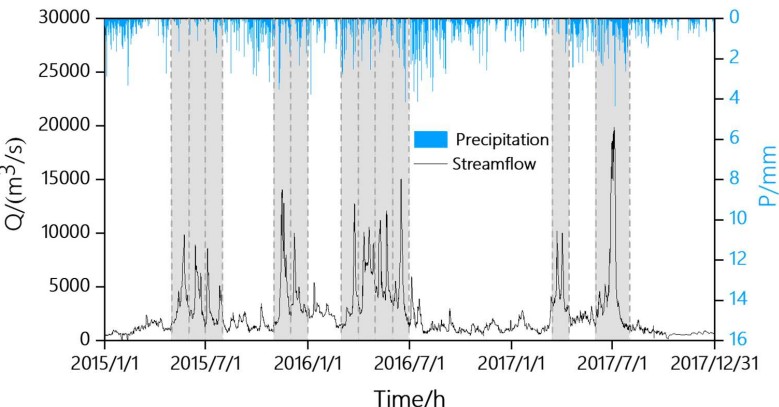


**Fig. 2. Time series of observed hourly streamflow in Xiangtan station and basin-average precipitation from CMA, with eleven selected flood events covered by shaded areas.**

**3. Methodology**

In this study, the IMERG precipitation products (IMERG-E, IMERG-L and IMERG-F) are assessed

against the reference precipitation, namely CMA, under different spatio-temporal resolutions. As

mentioned above, three widely used and typical physically-based models (lumped HBV model, semi-

distributed SWAT model, and distributed DHSVM model), and one data-driven model (LSTM), are

employed to probe the impacts of spatio-temporal resolutions of precipitation on flood events simulation.

To investigate the impacts of spatial resolutions of precipitation on flood simulation, precipitation

estimates with different spatial resolutions, which is obtained by inverse distance interpolation (Franke

1982), are used to force the selected models, which are SWAT and DHSVM, as well as LSTM. To study

the influence of temporal resolutions of precipitation on flood simulation, HBV, DHSVM and LSTM are

utilized, which forced by precipitation with different temporal resolutions. These four models are

calibrated with two calibration strategies to investigated the potential benefits of the calibration strategy

based on flood events. Finally, the performance of flood events simulation under different scenarios are

compared and discussed. The designed framework for this study is shown in a flowchart in Fig. 3.



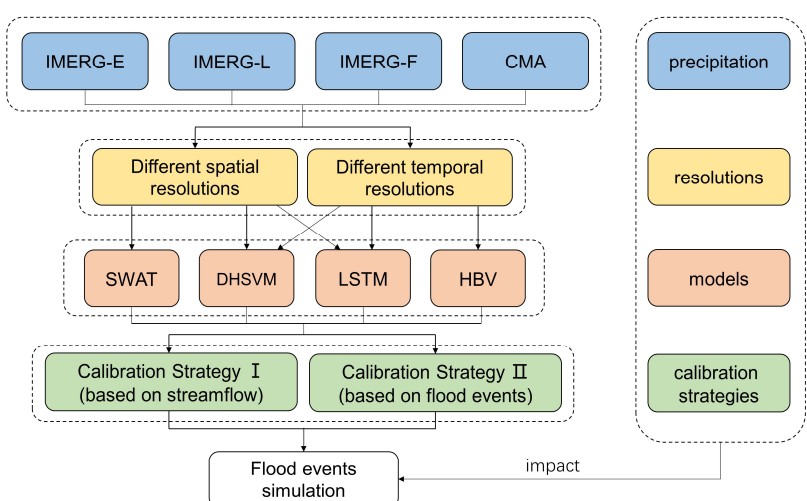

**Fig. 3. Methodological flowchart adopted in this study.**

## 3.1 Hydrological models and LSTM

### 3.1.1 The HBV model

The conceptual HBV model is originally developed by the Swedish Meteorological and Hydrological Institute (SMHI) (Bergström and Forsman 1973). Various versions of the HBV model have been developed and widely used in hydrological simulation and flood forecasting due to its simplicity and effectivity (Alfredsen and Hailegeorgis 2015, Grimaldi et al. 2019, Huang et al. 2019). A lumped version of HBV model is used in this study (AghaKouchak et al. 2013), which is operated at hourly and daily time steps with the inputs of precipitation, temperature and potential evapotranspiration. The potential evapotranspiration is calculated with the Penman-Monteith equation (Beven 1979) based on gauge meteorological data, and all the inputs are averaged over the basin with the Thiessen polygon method. Three main modules (soil moisture routine, response routine, and transformation routine) are contained in HBV model, while the module of snow routine is not included in this case because of the temperature above 0℃ perennially over the Xiang River basin.

### 3.1.2 The SWAT model

The SWAT model is a semi-distributed hydrological model developed by the Agricultural Research Center of the United States Department of Agriculture (USDA) (Arnold et al. 1998). The SWAT 2012 is



used in this study, which is operated on daily time step with the inputs of geographical data (DEM, land

use and soil) (Figure 4), precipitation and other meteorological variables mentioned above. The SWAT

model divides the watershed into sub-basins according to DEM, and then segregates them into multiple

hydrological response units (HRUs) as the basic computational unit based on different types of soil, land

use, and slope. The Xiang River Basin is divided into 25 sub-basins and 495 HRUs in this study. Forest-

evergreen is the dominant land cover category with coverage of 62%, and Ferralsols is main soil type

with coverage of 58%, as shown in the Fig. 4. The hydrologic cycle simulated by SWAT is based on the

water balance equation, which mainly includes surface runoff, evapotranspiration, soil moisture and

groundwater.

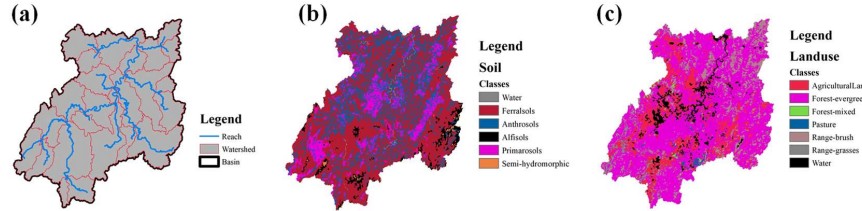

**Fig. 4. The (a) sub-basin divisions, (b) soil types, and (c) land use of Xiang River Basin used in SWAT model.**

**3.1.3 The DHSVM model**

The DHSVM model is a fully distributed, physics-based hydrological model developed by the Pacific

Northwest National Laboratory (PNNL) and the University of Washington (Wigmosta et al. 1994).

DHSVM uses near-surface meteorology including air temperature, wind speed, humidity, precipitation,

as well as incoming short- and long-wave radiation as hydro-climate inputs to solve energy and water

balance. The model represents a dynamic watershed processes at specific spatial scales considered the

effect of topography, soil, and vegetation. The DHSVM model mainly consists of seven modules,

including evapotranspiration module, surface snow melting module, canopy snow melting module,

unsaturated soil moisture module, saturated soil flow module, surface runoff module and flow routing

module. The version used in this study is DHSVM 3.1.2 with the grid resolution of 3,000 m. Six soil

types and eight vegetation classes are derived, and the spatial distributions of them are shown in the Fig.5.





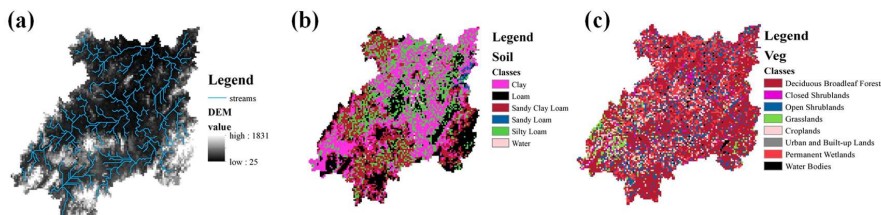

**Fig. 5. The (a) river network divisions, (b) soil types, and (c) vegetation types of Xiang River Basin used in DHSVM model.**

### 3.1.4 The Long Short-Term Memory network

The LSTM is a type of recurrent neural network (RNN), which is first proposed by Hochreiter and Schmidhuber (1997). LSTM is designed to overcome the error backflow problems with exploding and/or vanishing gradients by introducing three gates, namely forget, input, and output gates, into the repeating modules of neural network. The forget gate decides the information removed from the previous cell state. The input gate determines information updated to present cell state, and the output gate controls which part of the cell state output to the new hidden state. Therefore, LSTM can learn long-term dependencies between input and output features, which makes it appropriate for rainfall-runoff modeling. In this study, LSTM is developed using the deep learning framework PyTorch (Paszke et al. 2019), which has 100 hidden states and a single fully connected layer with a dropout rate of 0.5 (Srivastava et al. 2014). Precipitation and temperature are selected as the inputs of LSTM, and the output of LSTM is streamflow. The inputs for the complete sequence $x = [x1, ..., xn]$, where xt is a vector containing the input features of time t, and the dimension of the xt corresponds to the number of grids of the precipitation data. The outputs for the complete sequence $y = [y1, ..., yn]$, where yt is the streamflow of time t.

### 3.2 Two strategies for parameter calibration

### 3.2.1 Calibration Strategy I

As stated above, almost all the parameter calibration for hydrological modes is based on continuous streamflow, which is defined as Calibration Strategy I in this study. It is a conventional calibration method to optimize the parameters of hydrological model. For the HBV model, the whole period is divided into three periods: warm-up period (April 2014 to December 2014), calibration period (January 2015 to December 2016) and validation period (January 2017 to December 2017). The calibration is





conducted by maximizing the Nash–Sutcliffe efficiency coefficient (NSE) of the streamflow simulated during calibration period via the SCE-UA algorithm (Duan et al. 1994).

For the SWAT model, the whole period is also divided into three periods and they are the same as HBV. The calibration is accomplished with a separate tool named SWAT Calibration and Uncertainty Program

(SWAT-CUP) (Abbaspour et al. 2007). Parallel Sequential Uncertainty Fitting Version 2 (SUFI-2) is stable and always converging, and it is well appropriate for global optimization (Abbaspour et al. 2007), which is the reason why it is adopted in this study for parameter calibration. The objective function is also to reach the maximum value of NSE for the streamflow simulated in the calibration period.

The warm-up, calibration and validation periods of DHSVM are same as HBV and SWAT, as well as

the objective function. The parameter calibration of DHSVM is executed by an auto-calibration module based on ε-dominance non-dominated sorted genetic algorithm II (ε-NSGAII) (Pan et al. 2018). Parallel computing with a message passing interface (MPI) program is applied in this study.

Regarding the training of LSTM, the learnable parameters of the network are updated depending on a given loss function. Same as the selected hydrological models, the NSE is chosen as the objective

criterion for the LSTM (Kratzert et al. 2019), and adaptive moment estimation (Adam) (Kingma and Ba 2014) with the learning rate of 0.0001 is used as the optimization algorithm. The data set is divided into three parts generally, namely training, validation, and test data. The first two parts are used to determine the parameters of the networks, and the last one is used to evaluate the performance of actual application. In this study, the whole data set is divided into training set (October 2015 to December 2017) and

validation set (April 2014 to September 2015). The absence of test set is because of the limited available period of the data, while the selection of training period will be discussed in detail in section 5.3. Each LSTM networks are trained with three different random initial seeds for 1,500 epochs to account for the stochasticity in the network initialization. Among total 4,500 trained models, the best model is selected through comprehensive consideration of both calibration and validation NSE of the streamflow

simulation.

### 3.2.2 Calibration Strategy II

Calibration Strategy II is designed in this study particularly for flood events, which conducts the calibration based on the performance of flood event simulation. Eleven historical flood events occurred





between January 2015 and December 2017 are selected to conduct the flood events simulation (Fig. 2).

The calibration is conducted by maximizing the mean NSE of the flood events simulated during

calibration period for the HBV model. For the SWAT and DHSVM models, numerous sets of parameters

(the number is 1,000 in this study) are obtained through optimization algorithm, and the best fitted

parameters set is selected with the largest NSE for the flood events simulation. Considering the LSTM,

among total 4,500 trained models, the best model is also selected by maximizing the mean NSE of the

flood events simulation (4 flood events during calibration and 4 flood events during validation).

**3.3 Diagnostic statistics**

In order to quantitatively evaluate the performance of streamflow and flood events simulation, three

evaluation indices are selected in this study, namely NSE, BIAS-P and KGE. The formulas of these

indices are listed as follows:


$$NSE = 1 - \frac{\sum_{t=1}^{T} \left(Q_o^t - Q_s^t\right)^2}{\sum_{t=1}^{T} \left(Q_o^t - \overline{Q_o}\right)^2} \qquad (1)$$

$$BIAS - P = \left| \frac{Q_s^p - Q_o^p}{Q_o^p} \times 100 \right| \qquad (2)$$

$$KGE = 1 - \sqrt{(r-1)^2 + (\alpha-1)^2 + (\beta-1)^2} \qquad (3)$$

Where $Q_o^t$ and $Q_s^t$ are the values of the observed and simulated streamflow at time t; $Q_o^p$ and $Q_s^p$ are

the observed and simulated flood peak flows; r is the linear correlation between observations and

simulations, α a measure of the flow variability error, and β a bias term.

**4. Results**

**4.1 The performance of flood events simulation based on two different calibration strategies**

Fig. 6 shows the distributions of NSE and BIAS-P values, which are used to evaluate the performance of

four precipitation sources on flood events with two different calibration strategies at daily scale.



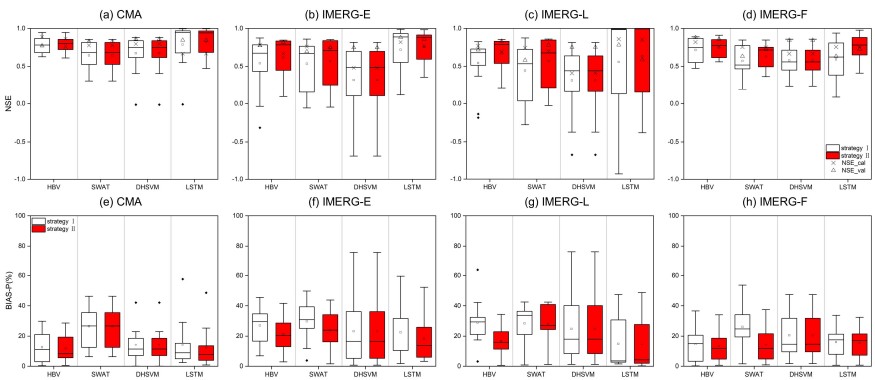

**Fig. 6. The NSE and BIAS-P of flood events simulation forced by (a, e) CMA, (b, f) IMERG-E, (c, g) IMERG-L and (d, h) IMERG-F using two calibration strategies (White box is based on calibration strategy I; red box is based on calibration strategy II). The box plots show the 25th, 50th, and 75th percentiles, and the mean value is given and shown by a square. The cross represents the NSE of simulated streamflow during calibration, and the triangle represents the NSE of simulated streamflow during validation.**

For the performance of HBV, it can be seen that flood events simulation with calibration strategy II shows better performance, for the mean NSE values of CMA, IMERG-E, IMERG-L, IMERG-F increase from 0.78, 0.54, 0.54, 0.72 with calibration strategy I to 0.79, 0.62, 0.67, 0.75 with calibration strategy II, respectively (Fig. 6). And the corresponding mean BIAS-P values decrease from 12.0%, 27.0%, 29.0%, 14.6% to 11.4%, 21.2%, 16.7%, 13.1%. Meanwhile, the uncertainty of NSE and BIAS-P values of flood events simulation is reduced, with less occurrences of poor flood events simulation. The flood events simulated by the CMA has the highest NSE among all precipitation sources, ranging from 0.61 to 0.95, and its averaged value is 0.79. It proves the capability of HBV in flood events simulation. When comparing the performance of IMERG precipitation estimates, the IMERG-F performs the best with both calibration strategies.

In terms of the streamflow and flood events simulation based on SWAT, Fig. 6 shows that the performance of the two calibration strategies with CMA is comparable, while for IMERG precipitation estimates, the strategy II outperforms the other one. Specifically, for streamflow simulation, the NSE values in the validation period of IMERG-E, IMERG-L, IMERG-F show a significant increase from 0.70, 0.58, 0.63 with the strategy I to 0.75, 0.78, 0.73 with the strategy II, respectively. For flood events simulation, the mean NSE values based on the strategy II are 0.57, 0.58, 0.63 forced with IMERG-E, IMERG-L, IMERG-F, which are 0.53, 0.44, 0.57 based on the strategy I. The corresponding mean BIAS-



P values are reduced from 29.8%, 28.4%, 26.1% to 23.9%, 28.0%, 13.2%. Compare to HBV and SWAT, the two calibration strategies present little difference in streamflow and flood events simulation based on

DHSVM, which indicates the performance of DHSVM is stable when using different calibration strategies.

For the LSTM, the NSE values of flood events simulation also show higher mean values and smaller uncertainty based on the strategy II for all precipitation products. The flood events simulation based on IMERG-F shows the most significant improvement with the mean NSE value increasing from 0.59 with

the strategy I to 0.75 with the strategy II. However, it also should be noted that despite calibration strategy II improves the flood events simulation forced by IMERG-L, the performance of LSTM on flood events simulation forced with IMERG-L is still unsatisfactory.

According to above results, it can be concluded that the calibration strategy II outperforms than the strategy I. Therefore, the following parts are based on the calibration strategy II.

**4.2 Impact of spatial resolutions of precipitation on flood events simulation**

To investigate the impact of spatial resolutions of precipitation on flood events simulation, the IMERG-E, IMERG-L, IMERG-F, and CMA are adopted to force the SWAT model, the DHSVM model and the LSTM model under 0.1°, 0.25° and 0.5°, respectively.

Fig. 7 shows the distributions of statistical indices, namely NSE, BIAS-P and KGE, which are used to

evaluate the performance of different precipitation sources with different spatial resolutions on flood events simulation. From the BIAS-P of flood events simulated with SWAT, it can be seen that spatial resolution significantly affects the performance of precipitation on flood events simulation. For instance, CMA performs the best at 0.25° with the mean BIAS-P of 26.5%, while IMERG-E, IMERG-L and IMERG-F display the best performance at 0.5° with the mean BIAS-P of 23.7%, 22.9% and 13.8%,

respectively. Similar to its performance in BIAS-P, in terms of NSE, CMA also performs the best under 0.25° with the mean NSE of 0.66. IMERG-E presents little difference at different spatial resolutions, while IMERG-L performs slightly better at 0.5° with the mean NSE of 0.61. The performance of IMERG-F gets worse as the resolution get coarser, regardless of the NSE or BIAS-P values. According to the KGE values, the performances based on CMA, IMERG-E and IMERG-L show improvement at coarser

spatial resolutions. Except for IMERG-F, whose KGE values are stable at 0.71.



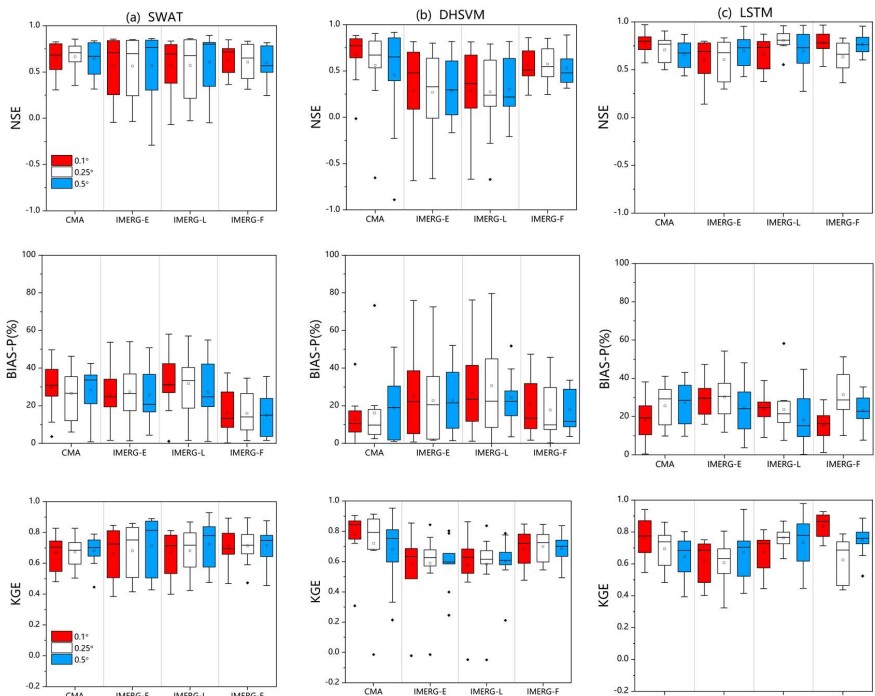

**Fig. 7. The performance of flood events simulation based on (a) SWAT, (b) DHSVM, and (c) LSTM forced by precipitation with different spatial resolutions. The box plots show the 25th, 50th, and 75th percentiles, and the mean value is given and shown by a square.**

Compare to SWAT, DHSVM shows different performance forced by precipitation with different spatial

resolutions. The mean NSE of flood events simulated with CMA declines from 0.68 to 0.45 when the

spatial resolution of precipitation changes from 0.1° to 0.5°, the mean KGE declines from 0.77 to 0.66,

and the mean BIAS-P increases from 13.9% to 19.3%. By contrast, the difference of flood events

simulated with IMERG forcing at different spatial resolutions is smaller, for instance, the mean NSE

values decrease from 0.32 to 0.30 for IMERG-E and IMERG-L, and 0.58 to 0.54 for IMERG-F. However,

the uncertainty of NSE, KGE and BIAS-P values of flood events simulated with IMERG is decreasing

as the spatial resolution. Among 11 flood events simulation, the performance of 4 flood events simulation

get better as the spatial resolution gets coarser. And the difference among the three IMERG precipitation

estimates is illustrated clearly in Fig. 7(b): the distribution of NSE, KGE and BIAS-P of simulated flood

events forced with IMERG-E is more scattered than the others, while the uncertainty of IMGER-F is the

smallest.



Similar to the performance of flood events simulated with SWAT, CMA performs the best at 0.25° with the mean BIAS-P of 13.8% and mean NSE of 0.84 based on LSTM. According to the BIAS-P values, when comparing the performance of the three IMERG precipitation estimates with different spatial

resolutions, all IMERG products perform the best at 0.1° with the mean BIAS-P of 18.4%, 14.2% and 15.5% for IMERG-E, IMERG-L and IMERG-F, respectively. In the light of NSE and KGE, IMERG-E and IMERG-F still achieve the best performance on flood events simulation at 0.1°, the mean values of which are 0.76, 0.59 and 0.75, 0.77, respectively. However, IMERG-L performs relatively poorly at 0.1°. Compared with the SWAT and DHSVM, the LSTM shows better performance on flood events simulation.

The mean NSEs of LSTM are higher than 0.7 in most instances, while the mean NSEs of SWAT is around 0.6, and the largest mean NSE of DHSVM is 0.68. The mean KGEs of SWAT and LSTM are similarly around 0.7, which are around 0.6 for DHSVM. In addition, LSTM also shows a relatively lower BIAS-P (the mean values less than 25%), except for the simulation based on IMERG-F under 0.5°.

**4.3 Impact of temporal resolutions of precipitation on flood events simulation**

To investigate the impact of temporal resolutions on flood events simulation, HBV, DHSVM and LSTM are adopted to be forced by the selected four precipitation sources at hourly and daily time scales. In order to compare the influences of temporal resolutions, the flood events simulated at hourly scale are aggregated into daily time series.

The performance of different precipitation datasets with different temporal resolutions on flood events

simulation is shown in Fig. 8. In HBV-based simulation, the mean NSE of the flood events simulation at the hourly scale is about 0.03 higher than that at the daily scale for all precipitation products. The mean KGE of the flood events simulation at the hourly scale is also higher than that at the daily scale for IMERG forcings, while the mean KGE of flood events simulated with CMA shows a decrease of about 0.03 at the hourly scale. In terms of BIAS-P, compared with the small difference between the

performance of flood events simulated with CMA at hourly and daily scales, the performance of IMERG-E, IMERG-L and IMERG-F on flood events simulation at hourly scale is much better than that at daily scale (mean BIAS-P values of 15.1% vs 21.2%, 13.7% vs 16.7% and 11.1% vs 13.1% for IMERG-E, IMERG-L and IMERG-F, respectively).



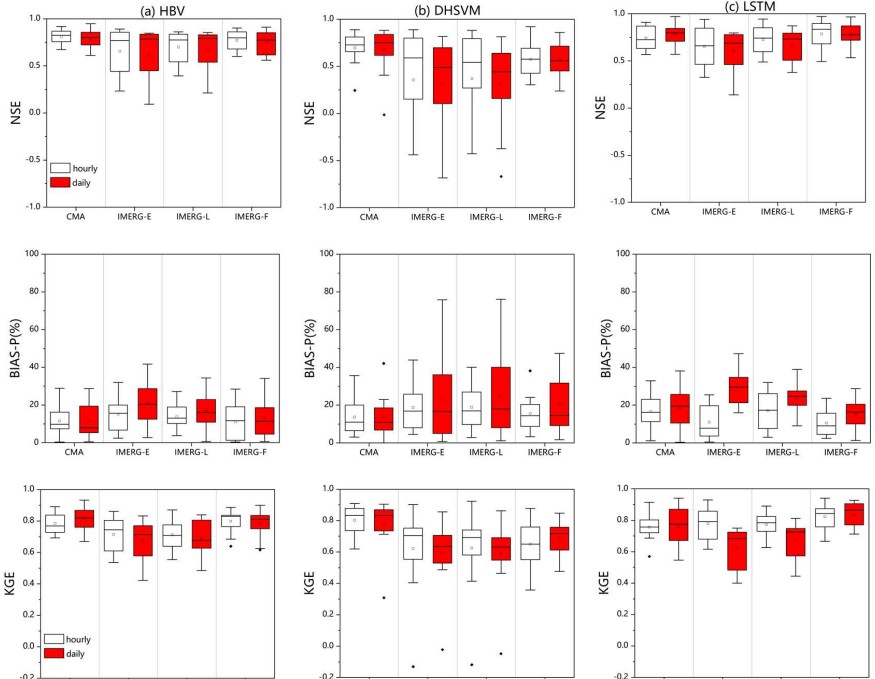

**Fig. 8 The performance of flood events simulation based on (a) HBV, (b) DHSVM, and (c) LSTM forced by precipitation with temporal resolution. The box plots show the 25th, 50th, and 75th percentiles, and the mean value is given and shown by a square.**

Similar performance is also presented in DHSVM-based simulation. According to the NSE and KGE values, the performances based on all precipitation products show improvement at the hourly scale. More

obvious improvement is shown in terms of BIAS-P, which is dropped by 5% at the hourly scale for IMERG products.

The performance based on LSTM is also shown in Fig. 8. Consistent with the results obtained by the HBV and DHSVM, all precipitation sources also have relatively better performance at the hourly scale. For example, the mean BIAS-P of CMA is reduced from 14.1% at the daily scale to 7.7% at the hourly

scale. And CMA, IMERG-L and IMERG-F obtain better performance at the hourly scale with the mean NSE of 0.83, 0.69 and 0.78, the mean KGE of 0.75, 0.74 and 0.79, respectively.

Compared with the HBV and DHSVM, the LSTM shows higher mean NSE values of flood events simulation, except for the simulation based on IMERG-L, while the HBV forced by CMA and IMERG-F presents smaller uncertainty. In terms of BIAS-P, two models show comparable performance with the





mean values around 15%. The performance on flood events simulation of HBV is more stable but slightly

poorer than LSTM in general.

## 5. Discussion

### 5.1 Comparison of two different calibration strategies

Two different calibration strategies are used to simulate flood events in this study. Compared with the

conventional method choosing the fit parameter set based on continuous streamflow (Calibration

Strategy I), selecting the parameter set that results in the best flood events simulation (Calibration

Strategy II) shows better performance on flood event simulation (Fig. 6). However, the CMA shows the

same results under two different calibration strategies in SWAT-based flood events simulation, so as the

DHSVM-based simulation. Furthermore, the CMA shows little difference with other precipitation

forcing. These findings indicate that both precipitation accuracy and calibration strategy used in

hydrological models are important uncertainty sources for flood simulation. From lumped model to

distributed model, precipitation accuracy becomes the major source of uncertainty to streamflow/flood

events simulation instead of hydrological model, the reason of which is that hydrological models describe

the hydrological process more and more comprehensively. In the application of LSTM for flood events

simulation, a large number of equivalent simulations with different parameter sets are generated, which

is similar to the parameter equifinality in hydrological simulation. When comparing the two calibration

strategies, the calibration strategy II is an effective way for training the LSTM model to obtain the best

flood events simulation results.

### 5.2 Comparison of the performance of precipitation products on flood events simulation at

**different spatio-temporal resolutions**

As illustrated in Fig.7 and Fig.8, the performance of precipitation products on flood events simulation is

affected by both the spatial and temporal resolutions.   Impacts of spatial resolution on flood events

simulation behave differently among different models and precipitation sources. For the study area, under

0.25° spatial resolution, the CMA obtains the best flood events simulation based on SWAT and LSTM.

The impact of spatial resolution on the capture of precipitation variability during flood event periods can

propagate to the flood events simulation. Best results are obtained under 0.25° spatial resolution, the


possible reason can be that finer spatial resolution (0.1°) increases the uncertainty of precipitation sets, nevertheless coarser spatial resolution (0.5°) decreases the sufficiency of datasets. It indicates that proper spatial resolution is essential to both minimize the uncertainty and assure the sufficiency.

The SWAT and DHSVM model driven by IMERG performs similarly under different spatial resolutions, which is consistent with previous research results (Lobligeois et al. 2014, Huang et al. 2019), where insignificant improvement was reported with higher spatial resolution of observed rainfall. It probably dues to the large catchment area and only the outlet station is used for calibration. Liang et al. (2004) found a critical resolution (1/8° for the VIC model) for a watershed with 1,233 km$^2$, beyond which the

spatial resolution shows limited impact on model performance. For our study area (82,375 km$^2$), when the spatial resolution of precipitation changed from 0.1° to 0.5°, small variety is shown in the performance of flood events simulation, which indicates the critical resolution may be larger for large watershed.

For data-driven model, IMERG-E and IMERG-F show better performance under 0.1° spatial resolution

in the LSTM-based simulation, which indicates that a higher spatial resolution (larger data set) can improve the performance of flood events simulation. Similar conclusion is drawn from previous study conducted by Sun et al. (2017), which also found that deep learning model performs better with larger datasets. In addition, the simulation with IMERG-L at 0.1° spatial resolution is not satisfactory, which may be related to the choice of hyperparameters and the limited data. However, after upscaling, the

performance of LSTM in flood events simulation is greatly improved when the IMERG-L data is applied with 0.25° spatial resolution, which implies that scale transformation can be regarded as an approach of data enhancement in hydrological simulation based on deep learning.

Considering the impacts of temporal resolutions on flood events simulation, for HBV and DHSVM, the flood events simulation at hourly scale outperforms than that at daily scale in general, which indicates

that a higher temporal resolution can improve the performance of hydrological models. Meanwhile, hourly precipitation sources also show better performance of flood events simulation with LSTM, especially for the simulation of flood peaks.





### 5.3 Comparison of different models on flood events simulation

In this study, a lumped hydrological model (HBV), a semi-distributed hydrological model (SWAT), a fully-distributed hydrological model (DHSVM) and a data-driven model (LSTM) are utilized to simulate flood events. In order to compare the performance of different models on flood events simulation more clearly, some results presented in Fig.6 are illustrated in Fig.9. As shown in Fig. 9, HBV and SWAT forced by CMA show comparable runoff simulation performance, while HBV shows better performance than SWAT in flood events simulation. The inability of the SWAT model to capture the flood events is

also proved in previous studies (Zhu et al. 2016, Yu et al. 2018). Furthermore, when driven by IMERG, HBV outperforms SWAT and DHSVM, especially by IMERG-E and IMERG-L, which is because the impact of error of precipitation is better constrained during its propagation in the hydrological process when utilizing hydrological model with simple structures (Zhu et al. 2013).




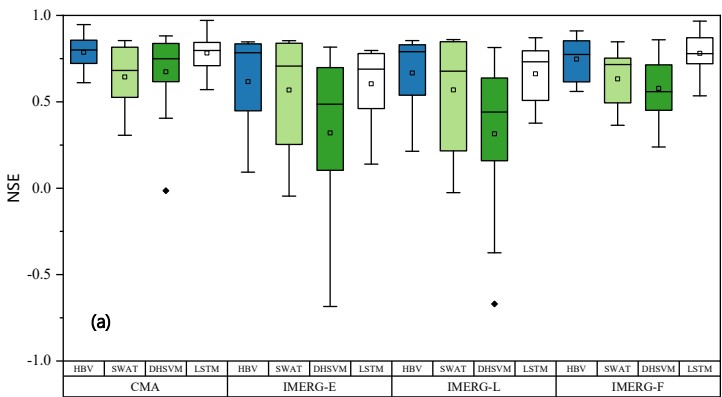

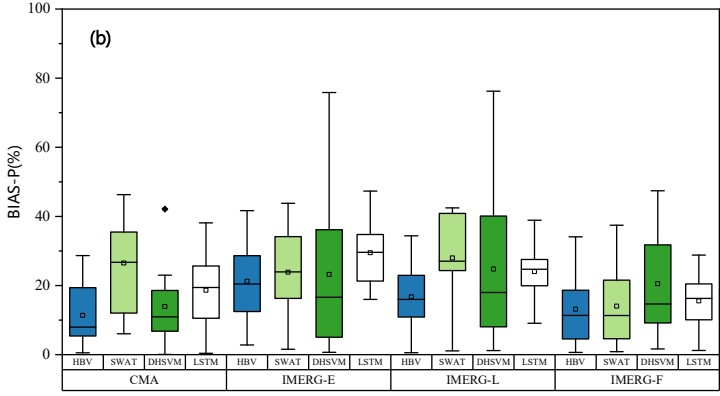

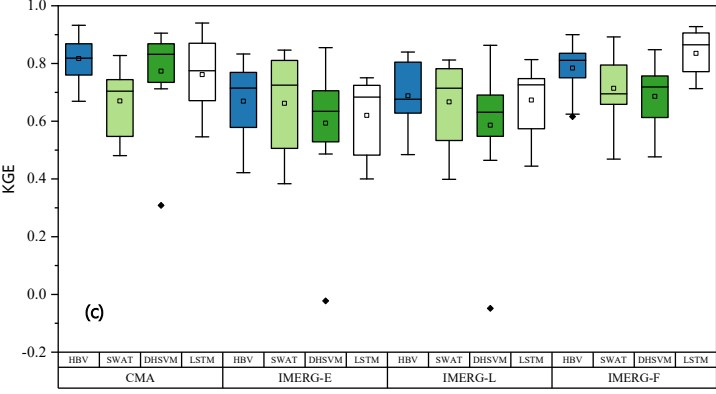

**Fig. 9. The (a) NSE , (b) BIAS-P and (c) KGE of flood events simulation forced by CMA, IMERG-E, IMERG-L and IMERG-F using calibration strategies II. The box plots show the 25th, 50th, and 75th percentiles, and the mean value is given and shown by a square.**




As a data-driven approach, LSTM shows better performance than hydrological models in terms of flood events simulation, which is considered an appropriate model in this case. Among IMERG products,

IMERG-F outperforms IMERG-E and IMERG-L in flood events simulation based on hydrological model, while IMERG-E and IMERG-L show comparable and even better performance than IMERG-F based on LSTM. This phenomenon shows that LSTM can deal with the error of precipitation products during the learning process. In many previous studies, LSTM is forced by large data set, such as the CAMELS data set, the lower bound of data requirements used for calibration is considered as the daily time series of 15

years (Kratzert et al. 2018, 2019). In this study, as mentioned above, the calibration (October 2015 to December 2017) and validation (April 2014 to September 2015) of LSTM are different from those of hydrological models. For hydrological models, the calibration period is from January 2015 to December 2016, and the calibration period is January 2017 to December 2017. We tried to use the same calibration data in LSTM as the hydrological model, but the results about flood events simulation is not satisfactory,

where its NSE of validation period is less than 0.5. The reason is that two major flood events are not included in the calibration period used in hydrological model. As a result, LSTM failed to learn the input–output relationship during the periods of flood events. Containing the characteristics of inputs as many as possible is critical for data-driven model, for instance LSTM, to capture the accurate relationship between the inputs and the output. Therefore, we use the data in the latter part for calibration, through

which the performance of LSTM is significantly improved. It should be notable that reliance on data may still be a potential barrier for LSTM in the data-sparse areas. In addition to obtaining more data for the input, such as the remote sensing data, how to make good use of limited data should also be considered in the future studies. What's more, based on the same computer specification (Intel i5-9300H CPU, 8 GB Memory), the running time of one simulation based on HBV, SWAT, DHSVM, and LSTM are 0.2

seconds, 1 minute, 54 minutes and 1.2 seconds, respectively. Results obtained from this case show that LSTM can provide reasonable accuracy in flood events simulation whilst it is also competitive in computational efficiency.

In order to compare the performance of flood events simulation with different scenarios, two randomly selected flood events simulation from July 1st, 2015 to July 31th, 2015, and from March 15th, 2017 to

April 14th, 2017 are shown in the Fig. 10. The first flood event is the typical one with single peak



occurred during the calibration period of HBV, SWAT and DHSVM models, and the latter one is with twin peaks occurred during the validation period of HBV, SWAT and DHSVM. While for LSTM, the occurrence times of the two selected flood events are in its validation and calibration period, respectively. From the figure, it can be seen that hydrological models generally show good capability to capture the

first flood event. However, for the second flood event from March 15th, 2017 to April 14th, 2017, an obvious underestimation of the first peak exists in the flood simulation, which is primarily caused by the bias of precipitation products, which are comprehensively evaluated in our previous study (Zhu et al. 2020a). The underestimation of the second flood peak is reduced in LSTM-based simulations, which implies the ability of LSTM to correct the propagation of influence from the bias of precipitation. Since

the hydrological models may smooth the short-term variability of input, the flood events simulated with hydrological models show relatively smooth runoff processes, compared with LSTM. Meanwhile, the performance of LSTM is not stable under different spatial resolutions, compared with the SWAT and DHSVM. Compared with spatio-temporal resolutions of precipitation and simulation models, precipitation source is the primary uncertainty source for flood events simulation, which indicates the

importance of choosing appropriate precipitation source for ungagged regions.



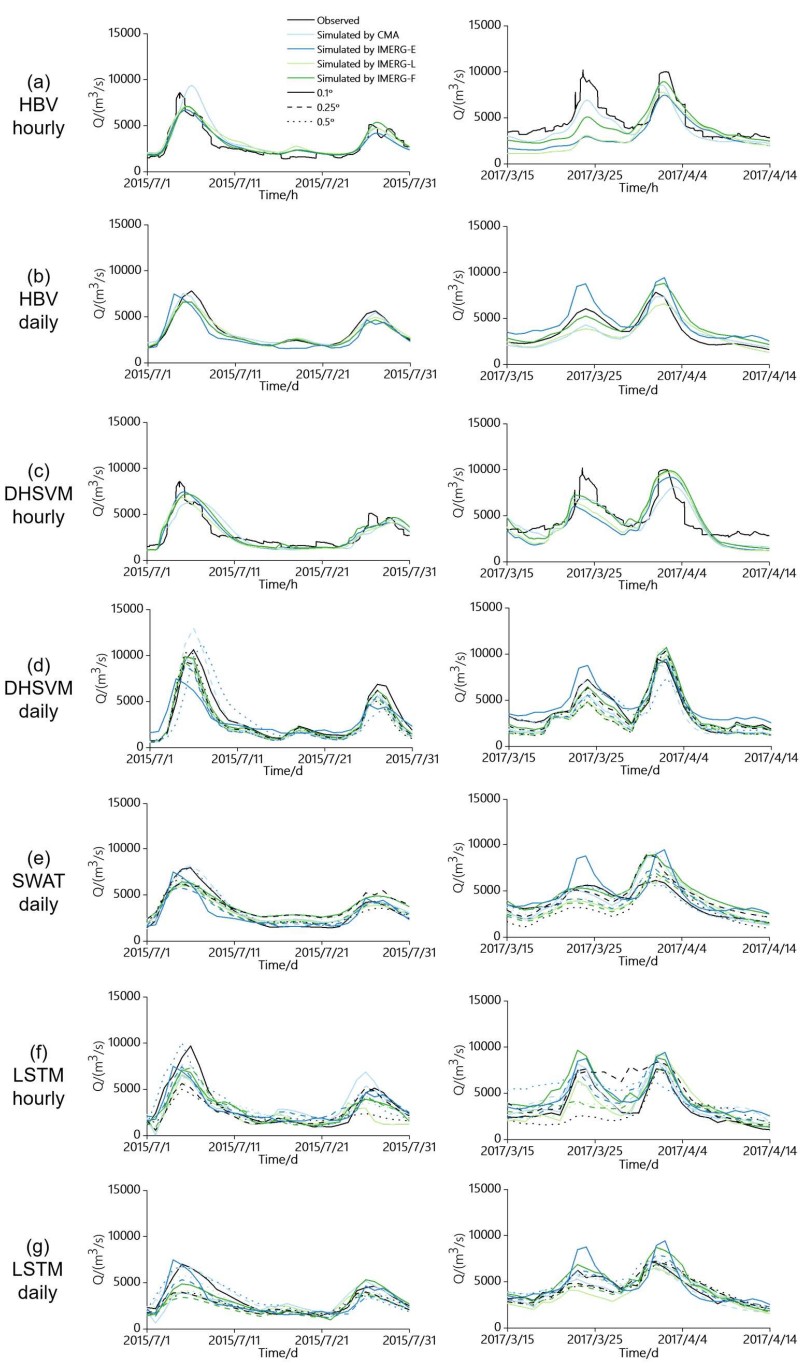

**Fig. 10. Comparison of HBV-, SWAT-, DHSVM-, and LSTM- based flood events simulation from July 1st, 2015 to July 31th, 2015, and from March 15th, 2017 to April 14th, 2017 forced by CMA, IMERG-E, IMERG-L, and IMERG-F with different spatio-temporal resolutions.**




## 6. Conclusion

In this study, we investigated the impacts of temporal and spatial resolutions of precipitation on flood events simulation over a large-scale catchment. We accomplished the study with the applicability of HBV, SWAT, DHSVM and LSTM forced by high spatio-temporal resolution gauge-based and satellite-based precipitation products. The main conclusions of this study are summarized as follows:

(1) According to the comparison of two calibration strategies, event-based calibration strategy leads to better performance of flood event simulation based on lumped HBV model and semi-distributed SWAT model. However, there is little difference between two calibration strategies application on distributed DHSVM model. For the data-driven model, LSTM, the event-based strategy also leads to better results.

(2) Considering the impact of temporal resolution, both hydrological models and LSTM perform better at hourly scale on flood events simulation than at daily scale, especially in flood peaks. However, the influence of spatial resolution on flood events simulation has no significant pattern in this case, which varies with models and precipitation sources.

(3) Three hydrological models and LSTM are used to simulate the flood events forced by gauge-based and satellite-based precipitation products in this study. The hydrological models and LSTM forced by IMERG precipitation estimates can achieve acceptable flood events simulation in most cases. In some cases, the LSTM outperforms the hydrological models. However, it should be notable that the performance of LSTM largely depends on the input data and settings such as the choice of hyperparameters, which may be unstable in some other cases.

### Acknowledgement

This study is financially supported by the National Natural Science Foundation of China (52009020) and the Natural Science Foundation of Jiangsu Province (BK20180403). This study is also financially supported by the High-level innovation and entrepreneurship talents plan of Jiangsu Province "Coupling remote sensing datasets to investigate impacts of hydrological key variables on flood extremes", and partially supported by the U.S. Department of Energy (DOE Prime Award # DE-IA0000018).





Appendix A



545

Fig. A0. Same as Fig. 7, but the results in calibration and validation periods are seperated



**Appendix B**

550

**Fig. B0. Same as Fig. 8, but the results in calibration and validation periods are seperated**





**Appendix C**

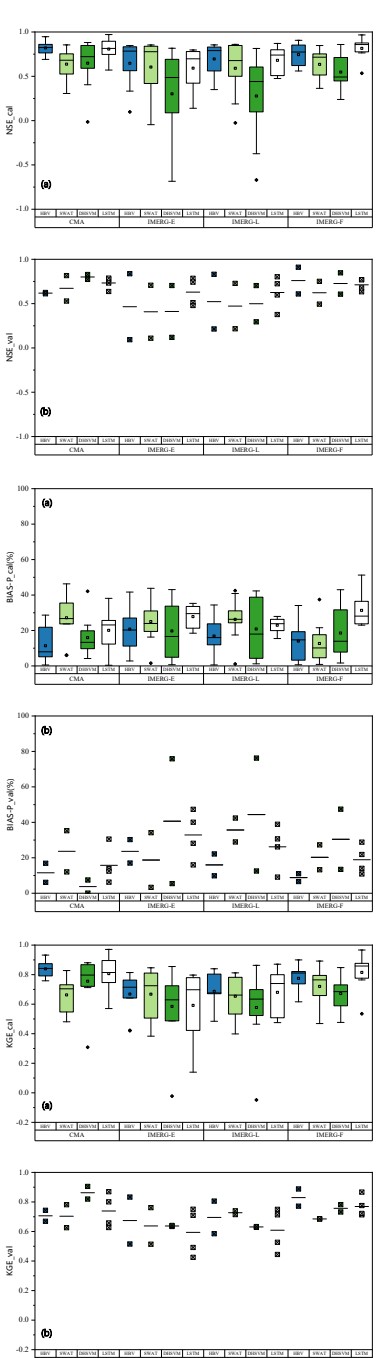

555

**Fig. C0. Same as Fig. 10, but the results in calibration and validation periods are seperated**



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
