# Peer review of "Impacts of spatio-temporal resolutions of precipitation on flood events simulation based on multi-model structures"

_Hydrology and Earth System Sciences, 2022_

## Author Comment (AC2)

Dear reviewer,

Thanks a lot for your great efforts to read through this paper and give very valuable comments. Here we have addressed the comments from you and the detailed description is attached in this document.

Best regards,

Qian Zhu, Xiaodong Qin, Dongyang Zhou, Tiantian Yang, Xinyi Song

**Point 1: Model calibration considering parts of discharge time series is not a new idea**

**Response 1:** Thank you very much for your comment. Input data, model and calibration strategy can affect the accuracy of flood events simulation and prediction. To our best knowledge, the sensitivity of models with different structures, such as lumped hydrological model, semi-distributed/distributed hydrological model, and data-driven model, to the spatio-temporal resolutions of precipitation has not been investigated. In this study, we investigated the impacts of temporal and spatial resolutions of precipitation on flood events simulation over a large-scale catchment, and we accomplished the study with the applicability of HBV, SWAT, DHSVM and LSTM forced by high spatio-temporal resolution gauge-based and satellite-based precipitation products.

**Point 2: Lines 20: It is not clear what you mean by "flood event." Also, I am not comfortable with the term "to match continuous streamflow." May be you can write "to match the entire streamflow time series."**

**Response 2:** Thank you very much for your comments. We have modified the relevant

description of flood event in Lines 20: "Two calibration strategies are carried out, one of which targets at matching the flood, and the other one is the conventional strategy to match the entire streamflow time series."

**Point 3: How did you select the flood events**

**Response 3:** Thank you for your question. ***In 2.2 Data description***, we have explained how we choose flood events: "Fig. 2 shows the time series of the hourly streamflow and corresponding gauge-based precipitation between 2015 and 2017, where eleven historical flood events are selected with flood peak exceeding the threshold of 8,600 m$^3$/s in this study."

[Figure]

**Fig. 2. Time series of observed hourly streamflow in Xiangtan station and basin-average precipitation from CMA, with eleven selected flood events covered by shaded areas.**

**Point 4: Line 295: Mean NSE may not be a reliable indicator. You should consider median, 75th and 25th percentile NSE.I see 75th NSE falling in case of CMA. The authors need to discuss it.**

**Response 4:**

Thank you for your suggestion. Since our target is to explore the impacts of different calibration strategies on flood events simulation, mean NSE is used in our study for it is more suitable for flood events as many previous studies proved (Yu et al. 2018, Kao et al. 2020). Meanwhile, the mean and median NSE have the same pattern in our study, the mean and median NSE of calibration strategy II are better than that of calibration strategy I as a whole, which is illustrated in Fig. 6, for HBV, the mean NSE values of CMA, IMERG-E, IMERG-L, IMERG-F increase from 0.78, 0.54, 0.54, 0.72 with calibration strategy I to 0.79, 0.62, 0.67, 0.75 with calibration strategy II, the median NSE increase from 0.78, 0.67, 0.79, 0.68 with calibration strategy I to 0.80, 0.78, 0.83, 0.79 with calibration strategy II.

**Point 5: NSEs in Figure 6: I don't see any consistent pattern. The results are not discussed properly.**

**Response 5:**

Thank you for your question, and sorry for the misunderstanding. In order to discuss the results more thoroughly, results and discussion are presented in two separate sessions. The mean and median NSE of calibration strategy II are better than that of calibration strategy I as a whole, which is illustrated in Fig. 6. For HBV, the mean NSE values of CMA, IMERG-E, IMERG-L, IMERG-F increase from 0.78, 0.54, 0.54, 0.72 with calibration strategy I to 0.79, 0.62, 0.67, 0.75 with calibration strategy II, the median NSE increase from 0.78, 0.67, 0.68, 0.79 with calibration strategy I to 0.80, 0.78, 0.79, 0.83 with calibration strategy II. For SWAT, the NSE values in the validation period of IMERG-E, IMERG-L, IMERG-F show a significant increase from 0.70, 0.58, 0.63 with the strategy I to 0.75, 0.78, 0.73 with the strategy II, the median NSE increase from 0.67, 0.53, 0.51 with the strategy I to 0.70, 0.67, 0.63 with the strategy II. For the LSTM, the NSE values of flood events simulation also show higher mean values and smaller uncertainty based on the strategy II for all precipitation products, the flood events simulation based on IMERG-F shows the most significant

improvement with the mean NSE value increasing from 0.59 with the strategy I to 0.75 with the strategy II, the median NSE value increase from 0.62 to 0.77.

[Figure]

**Fig. 6. The NSE and BIAS-P of flood events simulation forced by (a, e) CMA, (b, f) IMERG-E, (c, g) IMERG-L and (d, h) IMERG-F using two calibration strategies (White box is based on calibration strategy I; red box is based on calibration strategy II). The box plots show the 25th, 50th, and 75th percentiles, and the mean value is given and shown by a square. The cross represents the NSE of simulated streamflow during calibration, and the triangle represents the NSE of simulated streamflow during validation.**

**Point 6: NSEs in Figure 7: Again, I do not see a consistent pattern.**

**Response 6:** Thank you for your comment. In the original manuscript, we have discussed why there is not a consistent pattern for NSEs, and we think the impacts of spatial resolution on flood events simulation behave differently among different models and precipitation sources. The discussion part is as follows:

Page 19-20 Line 450-474 'For the study area, under 0.25° spatial resolution, the CMA obtains the best flood events simulation based on SWAT and LSTM. The impact of spatial resolution on the capture of precipitation variability during flood event periods can propagate to the flood events simulation. Best results are obtained under 0.25° spatial resolution, the possible reason can be that finer spatial resolution (0.1°) increases the uncertainty of precipitation sets, nevertheless coarser spatial resolution (0.5°)

decreases the sufficiency of datasets.

The SWAT and DHSVM model driven by IMERG performs similarly under different spatial resolutions, which is consistent with previous research results (Lobligeois et al. 2014, Huang et al. 2019), where insignificant improvement was reported with higher spatial resolution of observed rainfall. It probably dues to the large catchment area and only the outlet station is used for calibration. Liang et al. (2004) found a critical resolution (1/8° for the VIC model) for a watershed with 1,233 km2, beyond which the spatial resolution shows limited impact on model performance. For our study area (82,375 km2), when the spatial resolution of precipitation changed from 0.1° to 0.5°, small variety is shown in the performance of flood events simulation, which indicates the critical resolution may be larger for large watershed.

For data-driven model, IMERG-E and IMERG-F show better performance under 0.1° spatial resolution in the LSTM-based simulation, which indicates that a higher spatial resolution (larger data set) can improve the performance of flood events simulation. Similar conclusion is drawn from previous study conducted by Sun et al. (2017), which also found that deep learning model performs better with larger datasets. In addition, the simulation with IMERG-L at 0.1° spatial resolution is not satisfactory, which may be related to the choice of hyperparameters and the limited data. However, after upscaling, the performance of LSTM in flood events simulation is greatly improved when the IMERG-L data is applied with 0.25° spatial resolution, which implies that scale transformation can be regarded as an approach of data enhancement in hydrological simulation based on deep learning.'

Reference:

Huang, Y., Bárdossy, A. and Zhang, K. 2019. Sensitivity of hydrological models to temporal and spatial resolutions of rainfall data. Hydrology and Earth System Sciences, 23(6), 2647-2663.

Lobligeois, F., et al. 2014. When does higher spatial resolution rainfall information improve streamflow simulation? An evaluation using 3620 flood events. Hydrology and Earth System Sciences, 18(2), 575-594.

Liang, X., Guo, J. and Leung, L. R. 2004. Assessment of the effects of spatial resolutions on daily water flux simulations. Journal of Hydrology, 298(1-4), 287-310.

Sun, C., et al. 2017. Revisiting Unreasonable Effectiveness of Data in Deep Learning Era. 2017 Ieee International Conference on Computer Vision (Iccv), 843-852.

**Point 7: Results and discussions should be put together. It is difficult to follow discussion when results are not immediately available.**

**Response 7:** Thank you very much for your comments. We are very sorry for the difficulty in reading. As we mentioned above, we separate the results and discussion parts for the reason that we can discuss the results more thoroughly. In order to make it easy to follow, we have pointed out where to find the corresponding results, for example, "Compared with the conventional method choosing the fit parameter set based on entire streamflow time series (Calibration Strategy I), selecting the parameter set that results in the best flood events simulation (Calibration Strategy II) shows better performance on flood event simulation (Fig. 6)." Hope for your understanding.

---

## Author Response (AR4)

**Author's response**

Dear editor and reviewers,

Thanks a lot for your great efforts to read through this paper and give very valuable comments. Here we have addressed the comments from you and the detailed description is attached in this document.

Best regards,
Qian Zhu, Xiaodong Qin, Dongyang Zhou, Tiantian Yang, Xinyi Song

**Response to editor**

**Point 1: The comments of referees are very valid. The authors have shown that they appreciate these and have a plan for the paper revision. I would suggest to give special attention to the general comments of Referee 1, more clearly highlighting the novelty of this work.**

**Response 1:** Thank you very much for your comment. We have carefully responded to the comments of Referee 1 and the novelty of this work is clearly highlighted in the introduction part, which are listed as follows:

Page 3-4 Lines 85-94: 'But rare studies have been conducted to probe the effects of spatio-temporal of satellite-based precipitation on flood simulation, not to mention its impact on flood simulation with models based on DL methods (*e.g.*, LSTM). What's more important, to our best knowledge, the sensitivity of models with different structures, such as lumped hydrological model, semi-distributed/distributed hydrological model, and data-driven model, to the spatio-temporal resolutions of precipitation has not been investigated. Therefore, three widely used and typical physically based models (lumped HBV model, semi-distributed SWAT model, and distributed DHSVM model), and one data-driven model (LSTM) which shows good

performance in hydrological simulation, are employed to probe the impacts of spatio-temporal resolutions of precipitation on flood events simulation.'

Page 4 Lines 105-108: 'However, studies about event-based calibration are still quite limited, particularly for LSTM. Therefore, in this study, we conduct different calibration strategies aimed at obtaining the best possible flood events simulation.'

**Response to community comment**

**Point 1: Meaningful study! could the authors distinguish the sensitivities of these different models to the spatio-temporal resolutions of precipitation? and explain the reasons?**

**Response 1:** Thank you for your question. In our study, the hydrological models are more sensitive than the machine learning model on the whole, but the sensitivity of the model is related to the precipitation input.

As illustrated in Fig.7 and Fig.8, when the spatiotemporal resolution of CMA changes, DHSVM is the most sensitive one, the mean NSE of flood events simulated with which declines from 0.68 to 0.45 when the spatial resolution of precipitation changes from 0.1° to 0.5°. Perhaps, in the case of CMA-driven DHSVM, the impact of spatial resolution on the capture of precipitation variability during flood event periods can propagate to the flood events simulation.

But when the spatiotemporal resolution of IMERG changes, the SWAT and DHSVM model perform similarly under different spatial resolutions, which is consistent with previous research (Lobligeois et al. 2014, Huang et al. 2019), where insignificant improvement was reported with higher spatial resolution of observed rainfall in a large catchment area. It probably dues to the large catchment area and only the outlet station is used for calibration. Liang et al. (2004) found a critical resolution (1/8° for the VIC model) for a watershed with 1,233 km$^2$, beyond which the spatial resolution shows limited impact on model performance. For our study area (82,375 km$^2$), when the spatial resolution of precipitation changes from 0.1° to 0.5°, a small variation is shown in the performance of flood events simulation, which indicates the critical resolution may be larger for large watersheds. For HBV, it is not sensitive to changes in temporal resolutions because its simple hydrological model structure.

For LSTM, even though its sensitivity to the precipitation is lower than that of hydrological models, a higher resolution shows better performance. A similar

conclusion is drawn from a previous study conducted by Sun et al. (2017), which found that deep learning model performs better with larger datasets.

References:

Huang, Y., Bárdossy, A. and Zhang, K. 2019. Sensitivity of hydrological models to temporal and spatial resolutions of rainfall data. Hydrology and Earth System Sciences, 23(6), 2647-2663.

Lobligeois, F., et al. 2014. When does higher spatial resolution rainfall information improve streamflow simulation? An evaluation using 3620 flood events. Hydrology and Earth System Sciences, 18(2), 575-594.

Liang, X., Guo, J. and Leung, L. R. 2004. Assessment of the effects of spatial resolutions on daily water flux simulations. Journal of Hydrology, 298(1-4), 287-310.

Sun, C., et al. 2017. Revisiting Unreasonable Effectiveness of Data in Deep Learning Era. 2017 Ieee International Conference on Computer Vision (Iccv), 843-852.

**Response to Referee 1**

**Point 1: Model calibration considering parts of discharge time series is not a new idea**

**Response 1:** Thank you very much for your comment. We agree that model calibration considering parts of discharge time series is not a new idea for hydrological model. As we clarified in the introduction part, "However, studies about event-based calibration are still quite limited, particularly for LSTM. Therefore, in this study, we conduct different calibration strategies aimed at obtaining the best possible flood events simulation." Furthermore, besides calibration strategy, input data and model structure are the two main factors which affect the accuracy of flood events simulation and prediction, which are actually our primary focus. To our best knowledge, the sensitivity of models with different structures, such as lumped hydrological model, semi-distributed/distributed hydrological model, and data-driven model, to the spatio-temporal resolutions of precipitation has not been investigated. In this study, we investigated the impacts of temporal and spatial resolutions of precipitation on flood events simulation over a large-scale catchment, and we accomplished the study with the application of HBV, SWAT, DHSVM and LSTM forced by high spatio-temporal resolution gauge-based and satellite-based precipitation products.

**Point 2: Lines 20: It is not clear what you mean by "flood event." Also, I am not comfortable with the term "to match continuous streamflow." May be you can write "to match the entire streamflow time series."**

**Response 2:** Thank you very much for your comment. We have modified the relevant description of flood event in Lines 19-22: "Two calibration strategies are carried out, one of which targets at matching the flood events with peak discharge exceeding 8600

m$^3$/s between January 2015 and December 2017, and the other one is the conventional strategy to match the entire streamflow time series."

**Point 3: How did you select the flood events**

**Response 3:** Thank you for your question. ***In 2.2 Data description***, we have explained how we choose flood events: "Fig. 2 shows the time series of the hourly streamflow and corresponding gauge-based precipitation between 2015 and 2017, where eleven historical flood events are selected with flood peak exceeding the threshold of 8,600 m$^3$/s in this study."

[Figure]

**Fig. 2. Time series of observed hourly streamflow in Xiangtan station and basin-average precipitation from CMA, with eleven selected flood events covered by shaded areas.**

**Point 4: Line 295: Mean NSE may not be a reliable indicator. You should consider median, 75th and 25th percentile NSE. I see 75th NSE falling in case of CMA. The authors need to discuss it.**

**Response 4:** Thank you for your suggestion. Since our target is to explore the impacts of different calibration strategies on flood events simulation, mean NSE is used in our study for it is more suitable for flood events as many previous studies proved (Yu et al.

2018, Kao et al. 2020). Meanwhile, the mean and median NSE have the same pattern in our study, the mean and median NSE of calibration strategy II are better than that of calibration strategy I as a whole, which is illustrated in Fig. 6, for HBV, the mean NSE values of CMA, IMERG-E, IMERG-L, IMERG-F increase from 0.78, 0.54, 0.54, 0.72 with calibration strategy I to 0.79, 0.62, 0.67, 0.75 with calibration strategy II, while the median NSE increase from 0.78, 0.67, 0.79, 0.68 with calibration strategy I to 0.80, 0.78, 0.83, 0.79 with calibration strategy II.

As you said, the 75th percentile of NSE decreases in case of CMA. Upon checking the values, we found that it falls from 0.865 with calibration strategy I to 0.855 with calibration strategy II, indicating a very slight difference. Additionally, the other evaluation index, BIAS-P, shows better performance for calibration strategy II compared to calibration strategy I. Therefore, since it is targeted to compare the two calibration strategies, as a whole, we can summarize that calibration strategy II is better than calibration strategy I.

**Point 5: NSEs in Figure 6: I don't see any consistent pattern. The results are not discussed properly.**

**Response 5:** Thank you for your question, and sorry for the misunderstanding. In order to discuss the results more thoroughly, results and discussion are presented in two separate sessions. The mean and median NSE of calibration strategy II are better than that of calibration strategy I as a whole, which is illustrated in Fig. 6. For HBV, the mean NSE values of CMA, IMERG-E, IMERG-L, IMERG-F increase from 0.78, 0.54, 0.54, 0.72 with calibration strategy I to 0.79, 0.62, 0.67, 0.75 with calibration strategy II, the median NSE increase from 0.78, 0.67, 0.68, 0.79 with calibration strategy I to 0.80, 0.78, 0.79, 0.83 with calibration strategy II. For SWAT, the NSE values in the validation period of IMERG-E, IMERG-L, IMERG-F show a significant increase from 0.70, 0.58, 0.63 with the strategy I to 0.75, 0.78, 0.73 with the strategy II, the median NSE increase from 0.67, 0.53, 0.51 with the strategy I to 0.70, 0.67, 0.63 with the strategy II. For the LSTM, the NSE values of flood events simulation also show higher

mean values and smaller uncertainty based on the strategy II for all precipitation products, the flood events simulation based on IMERG-F shows the most significant improvement with the mean NSE value increasing from 0.59 with the strategy I to 0.75 with the strategy II, the median NSE value increase from 0.62 to 0.77.

[Figure]

**Fig. 6. The NSE and BIAS-P of flood events simulation forced by (a, e) CMA, (b, f) IMERG-E, (c, g) IMERG-L and (d, h) IMERG-F using two calibration strategies (White box is based on calibration strategy I; red box is based on calibration strategy II). The box plots show the 25th, 50th, and 75th percentiles, and the mean value is given and shown by a square. The cross represents the NSE of simulated streamflow during calibration, and the triangle represents the NSE of simulated streamflow during validation.**

Please refer to "5.1 Comparison of two different calibration strategies" for the corresponding discussion. Thanks.

**Point 6: NSEs in Figure 7: Again, I do not see a consistent pattern.**

**Response 6:** Thank you for your comment. In the manuscript, we have discussed why there is not a consistent pattern for NSEs, and the impacts of spatial resolution on flood events simulation behave differently among different models and precipitation sources. The discussion part is as follows:

Page 18-19 Line 426-450 'For the study area, under 0.25° spatial resolution, the CMA obtains the best flood events simulation based on SWAT and LSTM. The impact of spatial resolution on the capture of precipitation variability during flood event periods

can propagate to the flood events simulation. The best results are obtained under 0.25° spatial resolution, the possible reason can be that finer spatial resolution (0.1°) increases the uncertainty of precipitation sets, nevertheless coarser spatial resolution (0.5°) decreases the sufficiency of datasets.

The SWAT and DHSVM model driven by IMERG perform similarly under different spatial resolutions, which is consistent with previous research (Lobligeois et al. 2014, Huang et al. 2019), where insignificant improvement was reported with higher spatial resolution of observed rainfall in a large catchment area. It probably dues to the large catchment area and only the outlet station is used for calibration. Liang et al. (2004) found a critical resolution (1/8° for the VIC model) for a watershed with 1,233 km$^2$, beyond which the spatial resolution shows limited impact on model performance. For our study area (82,375 km$^2$), when the spatial resolution of precipitation changes from 0.1° to 0.5°, a small variation is shown in the performance of flood events simulation, which indicates the critical resolution may be larger for a large watershed.

For data-driven model, IMERG-E and IMERG-F show better performance under 0.1° spatial resolution in the LSTM-based simulation, which indicates that a higher spatial resolution, namely a larger dataset, can improve the performance of flood events simulation. Similar conclusion is drawn from previous study conducted by Sun et al. (2017), which also found that a deep learning model performs better with larger datasets. In addition, the simulation with IMERG-L at 0.1° spatial resolution is not satisfactory, which may be related to the choice of hyperparameters and the limited data. However, after upscaling, the performance of LSTM in flood events simulation is greatly improved when the IMERG-L data is applied with 0.25° spatial resolution, which implies that scale transformation can be regarded as an approach of data enhancement in hydrological simulation based on deep learning.'

References:

Huang, Y., Bárdossy, A. and Zhang, K. 2019. Sensitivity of hydrological models to temporal and spatial resolutions of rainfall data. Hydrology and Earth System Sciences, 23(6), 2647-2663.

Lobligeois, F., et al. 2014. When does higher spatial resolution rainfall information improve streamflow simulation? An evaluation using 3620 flood events. Hydrology and Earth System Sciences, 18(2), 575-594.

Liang, X., Guo, J. and Leung, L. R. 2004. Assessment of the effects of spatial resolutions on daily water flux simulations. Journal of Hydrology, 298(1-4), 287-310.

Sun, C., et al. 2017. Revisiting Unreasonable Effectiveness of Data in Deep Learning Era. 2017 Ieee International Conference on Computer Vision (Iccv), 843-852.

**Point 7: Results and discussions should be put together. It is difficult to follow discussion when results are not immediately available.**

**Response 7:** We sincerely apologize for any difficulties you may have experienced while reading. As we outlined previously, we elected to separate the results and discussion sections, enabling us to delve into a more comprehensive examination of our findings. To enhance clarity and facilitate comprehension, we have highlighted where to locate the relevant results, for instance, "Compared with the conventional method choosing the fit parameter set based on entire streamflow time series (Calibration Strategy I), selecting the parameter set that results in the best flood events simulation (Calibration Strategy II) shows better performance on flood event simulation (Fig. 6)." Hope for your understanding.

**Point 8: 3.2.2. It is not clear whether you have considered peak discharge only or all the data points of the flood hydrographs. If the former is true, the number of data points is very small for any meaningful calibration. The term 'flood event' has not been explicitly defined, which gives rise to additional confusion.**

**Response8:** Thank you very much for your comment. When we trained the models, we use all the data points of the flood hydrographs instead of just the peak discharges. For the term "flood event", *In 2.2 Data description*, we have explained how we choose flood events: "Fig. 2 shows the time series of the hourly streamflow and corresponding gauge-based precipitation between 2015 and 2017, where eleven historical flood events are selected with flood peak exceeding the threshold of 8,600 $m^3$/s in this study."

[Figure]

**Fig. 2. Time series of observed hourly streamflow in Xiangtan station and basin-average precipitation from CMA, with eleven selected flood events covered by shaded areas.**

**Point 9:** 3.3. It is not clear if Eq. (1) uses only flood peaks or all the data points in the time series for computing NSE. If the former is the case, the metric is not reliable since there are not many data points considered by the author.

**Response9:** Thank you very much for your comment. To quantitatively evaluate the performance of flood events simulation, three evaluation indices are selected in this study, namely NSE, BIAS-P and KGE. In Eq. (1) and Eq. (3), we used all the streamflow data points of 11 flood events and calculated the evaluation indices for each flood event separately.

We have modified the relevant description of Eq. (1) and Eq. (3) in Page 12 Lines 284-287: 'Where $Q_o^t$ and $Q_s^t$ are the values of the observed and simulated flood events at time $t$; $Q_o^p$ and $Q_s^p$ are the observed and simulated peaks of the flood events; $r$ is the linear correlation between observations and simulations, $\alpha$ a measure of the flow variability error, and $\beta$ a bias term.

**Point 10:** Eq. (2): Bias is not typically presented in this way. Again, how reliable is the equation when there are so few data points?

**Response10:** Thank you very much for your comment. To quantitatively evaluate the performance of the flood peaks simulation, BIAS-P is selected in this study instead of the common BIAS. BIAS-P provides a more comprehensive reflection of the errors between observed and simulated flood peaks.

**Point 11:** 4.1. The results are not very surprising since you have calibrated the model for flood peaks only. I would be surprised if you also show an improvement in overall NSE (i.e., NSE considering all the data points).

**Response11:** As we respond above, we use all the data points of the flood hydrographs to calibrate the models, and the best model is selected by maximizing the mean NSE of the flood events simulation. All trained models show an improvement in overall NSE, but the selected models are those that performs the best on flood events simulation.

**Point 12:** 4.2. The effects of precipitation data type on model performance are quite informative. However, no proper explanation is provided in the discussion section, which makes the analysis incomplete. Line 460 is unclear. What do you mean by error propagation? Please explain instead of merely citing another paper.

**Response12:** Sorry for the misunderstanding. The section 4.2 is about the impact of spatial resolutions of precipitation on flood events simulation, rather than the effects of precipitation data type on model performance. To investigate the impact of spatial resolutions of precipitation on flood events simulation, the IMERG-E, IMERG-L, IMERG-F, and CMA are adopted to force the SWAT model, the DHSVM model and the LSTM model under 0.1°, 0.25°and 0.5°.

The relevant discussion about the effects of precipitation data type on model performance is presented in ***5.2 "Comparison of the performance of precipitation products on flood events simulation at different spatio-temporal resolutions".***

We are very sorry for the difficulty in reading. The sentence in Line 460 has been re-edited:

Page 20 Lines 464-466: 'Furthermore, when driven by IMERG, HBV outperforms SWAT and DHSVM, especially by IMERG-E and IMERG-L. It is because the hydrological model with a simpler structure can reduce the impact of errors in radar rainfall estimation, which is better constrained during its propagation in the hydrological process (Zhu et al. 2013).'

References:

Zhu, D., Peng, D. Z., and Cluckie, I. D.: Statistical analysis of error propagation from radar rainfall to hydrological models, Hydrology and Earth System Sciences, 17, 1445-1453, 10.5194/hess-17-1445-2013, 2013.

**Point 13:** 4.3. The results look interesting. Again, no explanation is provided. For example, why does 0.25-degree data give the best 75th NSE and the worst 25th NSE for HBV (Figure 7a)?

**Response13:** Thank you very much for your comment. Fig. 7 show the performance of flood events simulation based on SWAT, DHSVM, and LSTM forced by precipitation with different spatial resolutions. But the HBV is not included in this part. Since our target is to explore the impacts of different resolutions on flood events simulation, we focus on the overall performance of the model, therefore, the mean NSE is used in our study for it is more suitable for flood events as many previous studies proved (Yu et al. 2018, Kao et al. 2020).

References:
Kao, I. F., Zhou, Y., Chang, L.-C., and Chang, F.-J.: Exploring a Long Short-Term Memory based Encoder-Decoder framework for multi-step-ahead flood forecasting, Journal of Hydrology, 583, 10.1016/j.jhydrol.2020.124631, 2020.
Yu, D., Xie, P., Dong, X., Hu, X., Liu, J., Li, Y., Peng, T., Ma, H., Wang, K., and Xu, S.: Improvement of the SWAT model for event-based flood simulation on a sub-daily timescale, Hydrology and Earth System Sciences, 22, 5001-5019, 10.5194/hess-22-5001-2018, 2018

**Point 14:** The abstract says LSTM is outperforming other models. Figure 6 says HBV is better than LSTM.

**Response14:** Thank you very much for your comment. Fig. 6 shows the distributions of NSE and BIAS-P values to illustrate the impact of calibration strategies on flood events

simulation. It can be seen that flood events simulation with LSTM shows better performance than HBV, for the mean NSE values of CMA, IMERG-E, IMERG-F increase from 0.79, 0.62, 0.75 based on HBV to 0.82, 0.76, 0.77 based on LSTM. For IMERG-L, the mean NSE of LSTM is slightly lower than HBV, but the BIAS-P of LSTM show better performance than HBV. As a whole, we can summarize that LSTM is better than HBV. And there is a section about the comparison of different models on flood events simulation in the discussion part. Please refer to "**5.3 Comparison of different models on flood events simulation**" for details.

**Point 15:** Line 90: The term 'physically based' is typically used for hydrological models based on hydrodynamic equations. The models you are referring to are typically called conceptual models. This is just a semantic issue though.

**Response15:** Thank you for pointing it out. Yes, HBV is a conceptual model, while SWAT and DHSVM are physically based models. And we revise the corresponding sentences as you suggested.

Page 2 Lines 38: 'Numerous models are applied to simulate the flood events, most of which are conceptual/physically based models.'

Page 4 Lines 91-94: 'Therefore, three widely used and typical conceptual/physically based models (lumped HBV model, semi-distributed SWAT model, and distributed DHSVM model), and one data-driven model (LSTM) which shows good performance in hydrological simulation, are employed to probe the impacts of spatio-temporal resolutions of precipitation on flood events simulation.'

Page 7 Lines 167-171: 'As mentioned above, three widely used and typical conceptual/physically based models (lumped HBV model, semi-distributed SWAT model, and distributed DHSVM model), and one data-driven model (LSTM), are employed to probe the impacts of spatio-temporal resolutions of precipitation on flood events simulation.'

**Point 16: Earlier I wrote "on. The term 'flood event' has not been explicitly defined." The concern remains unaddressed. The authors write "… 17, where eleven historical flood events are selected with flood peak exceeding the threshold of 8,600 m3/s in this study." Again, what is an event? Where does it start and where it ends? It appears the authors follow some subjective criteria to select the events, which are not elaborated.**

**Response16:** Thank you very much for your comment. The sentence has been re-edited: Page 5 Line 154-157 'Fig. 2 shows the time series of the hourly streamflow and corresponding gauge-based precipitation between 2015 and 2017, where eleven historical flood events are selected in this study. The flood events are the streamflow time series with one-month span whose peak flow exceeded $8600m^3/s$, corresponding to $97^{th}$ approximately the quantile level (Zhu et al., 2020a).'

References:

Zhu, Q., Zhou, D., Luo, Y., Xu, Y.-P., Wang, G., and Gao, X.: Suitability of high-temporal satellite-based precipitation products in flood simulation over a humid region of China, Hydrological Sciences Journal, 66, 104-117, 10.1080/02626667.2020.1844206, 2020a.

**Point 17: I am not comfortable with the overly simplistic conclusion that LSTM is better than HBV. The results provide a much more nuanced picture. Figure 6 NSE plots: HBV is more consistent across the data products compare to LSTM. For instance, LSTM's 25th percentile is much lower compared to HBV's for IMERG-L. A statement like "LSTM has a higher likelihood of success" would be much more acceptable.**

**Response17:**

Thank you very much for your suggestion and we agree with it. We add some discussion to compare the performance of LSTM and HBV in Figure 10 besides Figure 6, and the corresponding sentence has been re-edited:

Page 18-19 Line 535-538: 'The comparisons of SWAT, DHSVM and LSTM at different spatial resolutions are also illustrated. As a data-driven approach, LSTM shows better performance than SWAT and DHSVM in terms of flood events simulation and shows reduced uncertainty and a higher likelihood of success than HBV, which is considered an appropriate model in this case.'

**Point 18: I reiterate my earlier statement: there is no surprise that model performance improved for flood events after calibrating exclusively for flood events. It is widely acknowledged that calibration with respect to a specific objective function leads to its improvement. As I said earlier, I would be surprised to see overall NSE (NSE for the whole time series) improving after calibration with respect to flood events.**

**Response18:**

Thank you for this comment, and we calculate the NSE to see how the overall NSE performs for the whole time series. The results are presented in the following table. According to the Table 1, it can be seen that generally the calibration strategy II shows better performance than the strategy I in overall NSE, for the mean NSE values of HBV, SWAT, DHSVM, LSTM increase from 0.79, 0.77, 0.80, and 0.88 with calibration strategy I to 0.80, 0.80, 0.86, 0.89 with calibration strategy II. This is probably due to the fact that the NSE is more sensitive to changes in flood peaks (Huang et al., 2019) and calibration strategy II can better capture the flood peaks compared to calibration strategy I. Based on your comment, we have added some discussion in *"5.1 Comparison of two different calibration strategies"*, and the details are as follows:

 'Although we targeted in difference between the two strategies in flood events simulation, their performances in the whole streamflow simulation time series are also compared, which is presented in Table 1 (The mean value is the average NSE of the four precipitation products with the same calibration strategy). According to the mean NSE values, calibration strategy II outperforms calibration strategy I. To be specific, for HBV, SWAT, DHSVM and LSTM models, among the four precipitation products, there are two, three, three and three NSE values larger with calibration strategy II than that with calibration strategy I.'

**Table. 1 The NSE values of the whole streamflow simulation time series forced by CMA, IMERG-E, IMERG-L, IMERG-F**

| Model | Strategies | CMA | IMERG-E | IMERG-L | IMERG-F | Mean |
|-------|-----------|-----|---------|---------|---------|------|
| HBV | Strategies I | 0.77 | 0.77 | 0.72 | 0.88 | 0.79 |
| | Strategies II | 0.73 | 0.81 | 0.82 | 0.86 | 0.80 |
| SWAT | Strategies I | 0.83 | 0.75 | 0.76 | 0.73 | 0.77 |
| | Strategies II | 0.83 | 0.84 | 0.82 | 0.70 | 0.80 |
| DHSVM | Strategies I | 0.86 | 0.75 | 0.75 | 0.85 | 0.80 |
| | Strategies II | 0.82 | 0.87 | 0.86 | 0.87 | 0.86 |
| LSTM | Strategies I | 0.92 | 0.89 | 0.87 | 0.85 | 0.88 |
| | Strategies II | 0.93 | 0.91 | 0.86 | 0.85 | 0.89 |

References:

Huang, Y., Bárdossy, A., and Zhang, K.: Sensitivity of hydrological models to temporal and spatial resolutions of rainfall data, Hydrology and Earth System Sciences, 23, 2647-2663, 10.5194/hess-23-2647-2019, 2019.

**Point 19: As I mentioned earlier, there are numerous results but proportionally less discussion. For instance, no explanation is provided for why a 0.25-degree resolution appears to perform well for NSE-CMA-SWAT (apologies for the earlier typo) but not for NSE-CMA-LSTM. Similarly, a 0.5-degree resolution seems to work for KGE-CMA-SWAT but not for KGE-CMA-LSTM. The authors have predominantly presented the results without a thorough critical analysis, which is the point I am emphasizing. Once again, I am not suggesting that the study is irrelevant, but I believe that additional effort is needed to enhance the paper's overall appeal.**

**Response19:** Thank you for your comment, and we have added some results and discussion about this issue in **'5.2 Comparison of the performance of precipitation products on flood events simulation at different spatio-temporal resolutions'**:

Page 15-16 Line 478-491: 'In order to compare the performance of different models on flood events simulation in the same spatial resolutions, some results presented in Fig.7 are illustrated in Fig.9. Overall, the LSTM shows better performance in most cases, for instance, in Fig. 9 (a) and Fig. 9 (c), LSTM is better than other models with the largest mean NSE and the smallest range between $25^{th}$ and $75^{th}$ percentile. There is also exception, for example, in Fig. 9 (b), the range of NSE between $25^{th}$ and $75^{th}$ percentile of SWAT with CMA is smaller than that of LSTM, but its mean and medium values of NSE are lower. Therefore, it can be summarized that the performance of LSTM has a higher likelihood of success than the other models. For KGE at 0.1° (Fig.9 (d)), LSTM also show better performance than the other models expect that simulated with CMA, with which DHSVM is better than LSTM, and they show similar results with 0.5° (Fig. 9 (e)).'

[Figure]

**Fig. 9. The NSE and KGE of flood events simulation forced by CMA, IMERG-E, IMERG-L and IMERG-F with different spatial resolutions. The box plots show the $25^{th}$, $50^{th}$, and $75^{th}$ percentiles, and the mean value is given and shown by a square.**

As we stated in the manuscript, for the 0.25-degree resolution, it performs well for NSE-

CMA-SWAT and also for NSE-CMA-LSTM. Based on the mean and medium NSE values, NSE-CMA-LSTM is even better, with corresponding values increasing from 0.64 to 0.66.

To better explain the effect of spatial resolution on different models, we have added additional details in *'5.2 Comparison of the performance of precipitation products on flood events simulation at different spatio-temporal resolutions'*:

Page 16 Line 492-513: 'The influence of spatio-temporal resolution on flood events simulation is affected by model structure. For instance, based on NSE, the SWAT shows the best performance at 0.25° with CMA forcing, but the LSTM shows the best performance at 0.1°. Similarly, based on KGE, the SWAT performs the best at 0.5° with CMA forcing, but the LSTM has the best performance at 0.1°. On one hand, the difference in performance between NSE and KGE is due to their different statistical focus, with NSE giving larger weights to high values, especially flood peaks, which leads to different performance with different statistical metrics. On the other hand, the difference between SWAT and LSTM is due to their model structure. The SWAT operates as a physically driven model, where the impact of the spatial resolution of the precipitation dataset will propagate during hydrological process, which makes finer spatial resolution does not necessarily lead to the improved performance, as indicated by studies such as Huang et al. (2019). This is exemplified by the SWAT performs better at 0.25° with CMA forcing based on NSE, while it performs better at 0.5° based on

KGE. Regarding LSTM, as a deep learning model, some studies have highlighted significant performance enhancements when applied to larger, reliable datasets (Sun et al., 2017). Consequently, when forced by CMA and IMERG-F, LSTM shows the best performance across all statistical metrics at 0.1°, rather than at 0.25° or 0.5°. The deviations observed in IMERG-E and IMERG-L from this pattern are likely attributable to inherent errors within the precipitation product itself. We previously evaluated the applicability of the IMERG dataset in the Xiangjiang River Basin, and found that IMERG-E and IMERG-L have larger uncertainties and errors than IMERG-F (Zhu et al., 2020a). The CMA has been confirmed by several studies to be a more reliable

precipitation product in the Xiangjiang River Basin and always used as a reference precipitation product (Wang et al., 2017; Tang et al., 2017; Su et al., 2020). This probably makes IMERG-E and IMERG-L do not bring enough performance improvement to LSTM when the spatial resolution is finer.'

References:

Huang, Y., Bárdossy, A., and Zhang, K.: Sensitivity of hydrological models to temporal and spatial resolutions of rainfall data, Hydrology and Earth System Sciences, 23, 2647-2663, 10.5194/hess-23-2647-2019, 2019.

Su, J., Lü, H., Crow, W. T., Zhu, Y., and Cui, Y.: The Effect of Spatiotemporal Resolution Degradation on the Accuracy of IMERG Products over the Huai River Basin, Journal of Hydrometeorology, 21, 1073-1088, 10.1175/jhm-d-19-0158.1, 2020.

Sun, C., Shrivastava, A., Singh, S., and Gupta, A.: Revisiting Unreasonable Effectiveness of Data in Deep Learning Era, Ieee I Conf Comp Vis, 843-852, 10.1109/Iccv.2017.97, 2017.

Tang, G., Zeng, Z., Ma, M., Liu, R., Wen, Y., and Hong, Y.: Can Near-Real-Time Satellite Precipitation Products Capture Rainstorms and Guide Flood Warning for the 2016 Summer in South China?, IEEE Geoscience and Remote Sensing Letters, 14, 1208-1212, 10.1109/lgrs.2017.2702137, 2017.

Zhu, Q., Zhou, D., Luo, Y., Xu, Y.-P., Wang, G., and Gao, X.: Suitability of high-temporal satellite-based precipitation products in flood simulation over a humid region of China, Hydrological Sciences Journal, 66, 104-117, 10.1080/02626667.2020.1844206, 2020a.

Point 20: Mean NSE does not make a lot of sense if the distribution is skewed (which is very likely).

Response 20: Thank you very much for your comment. In most cases in our study, the medium NSE performs the same pattern as the mean NSE. For instance, in Fig. 6, the mean NSE values of HBV, SWAT, LSTM driven by IMERG-F increase from 0.54, 0.44, 0.56 with calibration strategy I to 0.67, 0.57, 0.63 with calibration strategy II while the medium NSE values increase from 0.68, 0.53, 0.98 to 0.79, 0.68, 0.99.

BIAS-P also shows the same pattern between the medium BIAS-P and the mean BIAS-P. For the mean BIAS-P values of HBV, SWAT, LSTM driven by IMERG-E decrease from 27.0%, 29.8%, 22.4% with calibration strategy I to 21.2%, 23.8%, 18.3% with calibration strategy II, while the medium BIAS-P values decrease from 34.3%, 37.6%, 27.2% to 27.8%, 32.5%, 23.1%. For the mean BIAS-P values of HBV, SWAT, LSTM driven by IMERG-F decrease from 14.5%, 26.0%, 16.0% with calibration strategy I to 13.1%, 14.0%, 15.5% with calibration strategy II while the medium BIAS-P values decrease from 15.0%, 24.7%, 17.5% to 11.3%, 11.3%, 13.1%.

Similar to Fig. 6, Fig. 7 and Fig. 8 show the same pattern between the medium NSE and the mean NSE in most case. For instance, in Fig. 7, the SWAT driven by CMA shows the best performance at 0.25° with the mean NSE of 0.66 and the medium NSE of 0.71. The SWAT driven by IMERG-E and IMERG-L show the best performance at 0.5° with the mean NSE of 0.57, 0.61 and the medium NSE of 0.76, 0.80. The SWAT driven by IMERG- F shows the best performance at 0.1°with the mean NSE of 0.63 and the medium NSE of 0.72. The LSTM driven by CMA and IMERG-F show the best performance at 0.1° with the mean NSE of 0.78, 0.78 and the medium NSE of 0.80, 0.78. The LSTM driven by IMERG-E shows the best performance at 0.5°with the mean NSE of 0.7 and the medium NSE of 0.73. The LSTM driven by IMERG-L shows the best performance at 0.25°with the mean NSE of 0.81 and the medium NSE of 0.81. In Fig. 8, the HBV driven by CMA and IMERG-F show the best performance at the hourly scale with the mean NSE of 0.81, 0.77 and the medium NSE of 0.82, 0.80. The DHSVM driven by IMERG-E and IMERG-L also show the best performance at the hourly scale with the mean NSE of 0.36, 0.37 and the medium NSE of 0.59, 0.54. So, in this study the medium NSE and the mean NSE are considered to be representative of the overall performance.

In some cases, as you said, mean NSE does not make a lot of sense if the distribution is skewed. For instance, in Fig. 6, the LSTM driven by CMA shows better medium NSE with calibration strategy I while better mean NSE with calibration strategy II. In Fig. 7, the DHSVM driven by IMERG-L shows the best medium NSE of 0.48 at 0.1° while the best mean NSE of 0.30 at 0.5.

In order to better describe the results, we use the 25th NSE and 75th NSE to discuss the uncertainty of the results when the distribution isskewed.

These sentences have been re-edited:

Page 10 Line 323-331: 'For the LSTM, the NSE values of flood events simulation also show higher mean values and smaller uncertainty based on the strategy II for all precipitation products. The flood events simulation based on IMERG-L shows the most significant improvement with the mean NSE value increasing from 0.62 with the strategy I to 0.77 with the strategy II. The flood events simulation based on CMA and IMERG-E show lightly lower medium NSE values of 0.94, 0.88 with the strategy II than 0.95, 0.99 with strategy I. But they show higher $25^{th}$ NSE with strategy II, especially LSTM driven by IMERG-E, which increases from 0.58 with strategy I to 0.66 with strategy II. Therefore, although strategy II has a lower median performance than strategy I in individual cases, it still significantly improves the performance of LSTM, particularly in terms of uncertainty.'

Page 10 Line 341-350: 'For instance, CMA performs the best at 0.25° with the mean BIAS-P of 26.5%, while IMERG-E, IMERG-L and IMERG-F display the best performance at 0.5° with the mean BIAS-P of 23.7%, 22.9% and 13.8%, respectively. Similar to its performance in BIAS-P, in terms of mean NSE, CMA also performs the best under 0.25° with the mean NSE of 0.66. IMERG-E presents little difference at different spatial resolutions, while IMERG-L performs slightly better at 0.5° with the mean NSE of 0.61and the medium NSE of 0.76.'

Page 11-12 Line 367-383: 'Similar to DHSVM, LSTM shows different performance forced by precipitation with different spatial resolutions. CMA and IMERG-F performs the best at 0.1° with the mean BIAS-P of 18.64% ,15.55% and mean NSE of 0.78. The $25^{th}$ NSE of flood events simulated with CMA increases from 0.52 to 0.72, the $75^{th}$ NSE increases from 0.78 to 0.83 while the spatial resolution is finer. By contrast, IMERG-E performs the best at 0.5° with the mean NSE of 0.69 and medium NSE of 0.68 while IMERG-L performs the best at 0.25° with the mean NSE of 0.80 and the medium NSE of 0.81. In the light of BIAS,

IMERG-E and IMERG-L achieve the best performance on flood events simulation at 0.5°, the mean values of which are 24.55%, and 18.27%, 0.77. In contrast of BIAS-P, LSTM driven by IMERG-L shows the best KGE at 0.25° with the mean KGE of 0.76 and the smallest uncertainty, which is the same as NSE. Compared with the SWAT and DHSVM, the LSTM shows better performance on flood events simulation. The mean NSEs of LSTM are higher than 0.7 in most cases, while the mean NSEs of SWAT is around 0.6, and the largest mean NSE of DHSVM is 0.68. The $25^{th}$ NSE of LSTM are higher than 0.5 in most cases, while the $25^{th}$ NSE of DHSVM is around 0.15. The smallest $75^{th}$ NSE of LSTM is 0.78, while the $75^{th}$ NSE of DHSVM are around 0.6. The mean KGEs of SWAT and LSTM are similarly around 0.7, which are around 0.6 for DHSVM. In addition, LSTM also shows a relatively lower BIAS-P (the mean values less than 25%).'

Page 14-15 Line 445-459: 'As illustrated in Fig.7 and Fig.8, the performance of precipitation products on flood events simulation is affected by both the spatial and temporal resolutions. Impacts of spatial resolution on flood events simulation behave differently among different models and precipitation sources. For the study area, under 0.25° spatial resolution, the CMA obtains the best flood events simulation based on SWAT. The impact of spatial resolution on the capture of precipitation variability during flood event periods can propagate to the flood events simulation. The best results are obtained under 0.25° spatial resolution, the possible reason can be that finer spatial resolution (0.1°) increases the uncertainty of precipitation sets, nevertheless coarser spatial resolution (0.5°) decreases the sufficiency of datasets. For SWAT driven by CMA, it shows the best $75^{th}$ NSE and the worst $25^{th}$ NSE at 0.5° while the DHSVM driven by CMA shows the same pattern at 0.5°, which proved that coarser spatial resolution decreases the sufficiency of datasets. But the DHSVM driven by CMA shows the best performance at 0.1°, which proves that the effects of increasing and decreasing spatial resolution are simultaneous and affect different models differently. It indicates that the choice of the dataset is influenced by the resolution range, which must be adapted to the model definition, for the proper spatial resolution is essential to both minimize the

uncertainty and assure the sufficiency (Grusson et al., 2017).'

References:

Grusson, Y., Anctil, F., Sauvage, S., and Sánchez Pérez, J.: Testing the SWAT Model with Gridded Weather Data of Different Spatial Resolutions, Water, 9, 10.3390/w9010054, 2017.

**Response to Referee 2**

**Point 1: Line 66: "the study"-> "they".**

**Response 1:** Thank you for your suggestion. This sentence has been re-edited:

Page 3 Lines 65-68: 'Su et al. (2020) assessed the IMERG products at multiple spatial and temporal resolutions by upscaling, and they summarized that degrading the spatio-temporal resolution improves the accuracy of IMERG products.'

**Point 2: Line 124-127: The structure of these two sentences is suggested to be revised. The conjunction "so" in the beginning of the second sentence may be unclear.**

**Response 2:** Thank you for your suggestion. This sentence has been re-edited:

Page 5 Lines 126-129: 'Concentrated storm events during the flood season cause frequent floods throughout the basin. Since the Xiang River basin is the most densely populated and economically developed area in Hunan Province (Zhu et al. 2020a), it is critical to accurately simulate and predict flood events in the region for effective flood risk management.'

**Point 3: Line 139: "(hereafter CMA)" needs to be put behind "China Meteorological Administration".**

**Response 3:** Thank you for your suggestion. This sentence has been re-edited:

Page 6 Lines 141: 'A precipitation product released by China Meteorological Administration (hereafter CMA),'

**Point 4: Line 185: the reference "(AghaKouchak et al. 2013)" should be located behind the "HBV model".**

**Response 4:** Thank you for your suggestion. This sentence has been re-edited:

Page 8 Lines 186-187: 'A lumped version of HBV model (AghaKouchak et al. 2013) is used in this study,'

**Point 5: Line 230-233: please pay attention to the format of the variables, such as xt, and t.**

**Response 5:** Thank you for your question. I am sorry for our carelessness; the format of the variables has been corrected:

Page 10 Lines 233-236: 'The inputs for the complete sequence $x = [x_1, ..., x_n]$, where $x_t$ is a vector containing the input features of time $t$, and the dimension of the $x_t$ corresponds to the number of grids of the precipitation data. The outputs for the complete sequence $y = [y_1, ..., y_n]$, where $y_t$ is the streamflow of time $t$.'

**Point 6: Line 268-269: please explain how the eleven historical flood events are selected.**

**Response 6:** Thank you for your question. ***In 2.2 Data description***, we have explained how we choose flood events: "Fig. 2 shows the time series of the hourly streamflow and corresponding gauge-based precipitation between 2015 and 2017, where eleven historical flood events are selected with flood peak exceeding the threshold of 8,600 m$^3$/s in this study."

[Figure]

**Fig.2. Time series of observed hourly streamflow in Xiangtan station and basin-average precipitation from CMA, with eleven selected flood events covered by shaded areas.**

**Point 7: Line 338: "as the resolution get coarser"-> as the resolution is coarser or as the resolution gets coarser.**

**Response 7:** Thank you for your suggestion. This sentence has been re-edited:

Page 14 Lines 340-341: 'The performance of IMERG-F gets worse as the resolution is coarser,'

**Point 8: Line 350-352: "However, the uncertainty of NSE, KGE and BIAS-P values of flood events simulated with IMERG is decreasing as the spatial resolution." As the spatial resolution what? finer or coarser?**

**Response 8:** Thank you for your question. We are very sorry for the difficulty in reading. This sentence has been re-edited:

Page 15 Lines 354-355: 'However, the uncertainty of NSE, KGE and BIAS-P values of flood events simulated with IMERG decreases as the spatial resolution is finer.'

**Point 9: Line 365: in most instances -> in most cases.**

**Response 9:** Thank you for your suggestion. This sentence has been re-edited:

Page 16 Lines 368: 'The mean NSEs of LSTM are higher than 0.7 in most cases,'

**Point 10: Line 407-408: "the same results" means the results are exactly the same, does that what the authors indicate? Otherwise, the same results -> the comparable/similar results or the results are almost the same.**

**Response 10:** Thank you for your suggestion. This sentence has been re-edited:

Page 18 Lines 410-412: 'However, the CMA shows the similar results under two different calibration strategies in SWAT-based flood events simulation.'

**Point 11: Line 417-418: the calibration strategy II is an effective way for training the LSTM model to obtain the best flood events simulation results -> the calibration strategy II is an effective way to train the LSTM model to obtain the best flood events simulation.**

**Response 11:** Thank you for your suggestion. This sentence has been re-edited:

Page 18 Lines 419-421: 'When comparing the two calibration strategies, the calibration strategy II is an effective way to train the LSTM model to obtain the best flood events simulation.'

**Point 12: Line 430: performs -> perform.**

**Response 12:** Thank you for your suggestion. This sentence has been re-edited:

Page 19 Lines 433: 'The SWAT and DHSVM model driven by IMERG perform similarly under different spatial resolutions,'

**Point 13: Line 431: please delete the "results". And please check the whole manuscript for this issue.**

**Response 13:** Thank you for your suggestion. This sentence has been re-edited:

Page 19 Lines 434: 'which is consistent with previous research (Lobligeois et al. 2014, Huang et al. 2019),'

And we checked the whole manuscript for this issue as you suggested. Thanks.

**Point 14: Line 440: larger data set -> larger dataset. Isn't the "Fig. 9" shall be colored red to be consistent with other figures?**

**Response 14:** Thank you for your suggestions. This sentence has been re-edited:

Page 19 Lines 443-444: 'which indicates that a higher spatial resolution, namely a larger dataset, can improve the performance of flood events simulation.'

We have changed the color of Fig. 9 to make it consistent with other figures.:

[Figure]

**Fig. 9. The (a) NSE, (b) BIAS-P and (c) KGE of flood events simulation forced by CMA, IMERG-E, IMERG-L and IMERG-F using calibration strategies II. The box plots show the 25th, 50th, and 75th percentiles, and the mean value is given and shown by a square.**

**Point 15: The colors used in Fig.10 are not so easy to distinguish.**

**Response 15:** Thank you for your question. We are very sorry for the difficulty in reading. We have changed the color of Fig.10 to make it easier to distinguish:

[Figure]

Fig. 10. Comparison of HBV, SWAT, DHSVM, and LSTM based flood events simulation from July 1st, 2015 to July 31th, 2015, and from March 15th, 2017 to April 14th, 2017 forced by CMA, IMERG-E, IMERG-L, and IMERG-F with different spatio-temporal resolutions.

**Point 16: Same issue of Appendix C, and please refer to the comment #15**

**Response 16:** Thank you for your question. We are very sorry for the difficulty in reading. We have changed the color of Appendix C to make it consistent with other figures.:

Fig. C0. Same as Fig. 9, but the results in calibration and validation periods are separated